

# Trace elements in mussel shells from the Brazos River, Texas: environmental and biological control

**Authors:** Alexander A. VanPlantinga[1] and Ethan L. Grossman[1]

[1]Department of Geology and Geophysics, Texas A&M University, College Station, Texas, USA 77843-3115

**Abstract.** In sclerochronology, understanding the drivers of shell chemistry is necessary in order to use shells to reconstruct environmental conditions. We measured the Mg, Ca, Sr, Ba, and Mn contents in water samples and in the shells of two freshwater mussels (*Amblema plicata* and *Cyrtonaias tampicoensis)* from the Brazos River, Texas to test their reliability as environmental archives. Shells were analyzed along growth increments using age models established with stable and clumped isotopes. Shells were also examined with cathodoluminescence (CL) microscopy to map Mn/Ca distribution patterns. Sr/Ca correlated with Mn/Ca, while Mg/Ca and Ba/Ca showed no clear trends. Mn/Ca correlated inversely with the log of river discharge. Because dissolved and inorganic particulate sources of manganese are low during low flow, peak Mn/Ca values may come from elevated feeding or metabolic rates. Shell Mn/Ca values were used to reconstruct river discharge patterns, which, to our knowledge, has previously only been performed with shell chemistry using oxygen isotopes.

## 1 Introduction

Sclerochronology is the study of the physical and chemical properties of invertebrate hard parts. There is great potential for using mollusks to reconstruct environmental conditions in the present and in the geologic past, but problems remain in understanding the relationship between mollusk shell chemistry and the ambient environment (Immenhauser et al., 2016). For example, shell Sr/Ca can record temperature as a reflection of mollusk metabolic response to seasonal temperature variation opposite what is thermodynamically predicted for aragonite (Wheeler, 1992; Gillikin et al., 2005; Carré et al., 2006; Sosdian et al., 2006; Gentry et al., 2008). Shell Mg/Ca can record temperature (Freitas et al., 2006), and shell Ba/Ca sometimes correlates with diatom primary productivity (Vander Putten et al., 2000; Lazareth et al., 2003), but it can also be controlled by growth rate (Izumida et al., 2011). Mollusk soft tissue reflects variations in metal bioaccumulation by organ and by element (Arafin and Bendell Young, 2000; Chale, 2002; Ravera et al., 2003; Silva et al., 2006; Bellotto and Mieckeley, 2007). Soft tissue bioaccumulation can in turn elucidate pathways to shell bioaccumulation (Puente et al., 1996; Bilos et al, 1998; Langlet et al., 2007).

Untangling the physical, chemical, and biological factors involved in sclerochronology will improve the utility of
mollusk shells as environmental archives (Vander Putten et al., 2000).

Studies of mollusk shell Mn/Ca have highlighted chemical, physical, and biological pathways of

environmental manganese, providing insight into mollusk physiology, ecosystems, food webs, and human impacts
such as soil erosion, eutrophication, and hypoxia (Risk et al., 2010; Langlet et al., 2007; Jacob et al., 2008; Zhao et
al., 2016; Zhao et al., 2017a). Aquatic manganese, whether dissolved or particulate, is controlled by redox
conditions (pH and DO), which are in turn controlled by nutrient flux (Langlet et al., 2007), microbial oxidation
(Sunda and Huntsman, 1990), and physical factors such as wind and water currents and photoreduction (Sunda and
Huntsman, 1994). Manganese can be incorporated in mollusk shells via suspended organic particle ingestion (Bilos
et al., 1998; Vander Putten et al., 2000; Lazareth et al., 2003; Langlet et al., 2007). Dissolved $Mn^{2+}$ is the most
bioavailable form of manganese (Campbell, 1995), and experimental studies using Mn-spiked water have shown the
direct influence of dissolved Mn on shell Mn (Jeffree et al., 1995; Hawkes et al., 1996;  Markich et al., 2002;
Langlet et al., 2006; Lartaud et al., 2010). Naturally dissolved Mn variation has been demonstrated to influence shell
Mn/Ca in several studies (Frietas et al., 2006; Barats et al., 2008; Zhao et al., 2017a). Nevertheless, little is known
about the spatial and temporal variation of dissolved and inorganic and organic forms of manganese, including the
chemistry of river colloids, sediment porewater, and phytoplankton.

While trace element studies of marine bivalves are common, trace element studies of freshwater mussels

are uncommon despite the fact that freshwater mussels are threatened worldwide by anthropogenic nutrient influxes
and water impoundment (Lydeard et al., 2004; Richter et al., 1997). Studies of freshwater mussel trace elements
have highlighted the relationship between shell metal/Ca (Me/Ca) values and water Me/Ca values (Carroll and
Romanek, 2008; Bolotov et al., 2015; Geeza et al., 2018), and relationships between Me/Ca and physical factors
such as river discharge (Risk et al., 2010) and nutrient pollution (Zhao et al., 2017a).

This study explores relationships between the Brazos River physical and chemical parameters and the Mg,

Sr, Ba, and Mn contents of freshwater mussel shells during the drought period of 2013. This work utilizes the
oxygen isotope sclerochronology from VanPlantinga and Grossman (2018) established with the aid of clumped
isotopes. This approach allows for the study of a challenging and dynamic environment, a subtropical regulated river
where the mussel shell isotope record cannot be tied to seasonal patterns as easily as in temperate, tropical, or



marine environments. Building on the water and shell isotope data, the present study focuses on trace metals and
their relation to river nutrients. Although the shell Sr/Ca-temperature relationship was expected, the inverse Mn/Ca-
discharge relationship indicates that river flow controls the bioavailability of manganese. Below we explore the
basis for this observation and recommend further research on river manganese flux.
**2 Methods**
**2.1 Setting, water sampling and analysis**

This study focuses on the middle run of the Brazos River near College Station, Texas (near the USGS gage

08108700 in Bryan, Texas) about 210 km north of its estuary in the Gulf of Mexico (Figure 1). Water impoundment
near this study site negatively impacts mussel diversity (Randklev et al., 2013; Tsakiris and Randklev, 2016) in the
Brazos River. The Brazos flows southeast through a semi-arid to semi-humid climate characterized by hot summers
and mild winters, averaging 29˚C and 13˚C, respectively (Nielsen-Gammon, 2011). Average annual rainfall in
College Station is 100 cm and historically peaks in late-spring and mid-fall. About 240 km upstream of the study
site is Lake Whitney, dammed for hydropower and flood control. About 30 km upstream of the study site is the
confluence with the Little River, the largest Brazos tributary, receiving flows from Lake Belton, Stillhouse Hollow
Lake, and Granger Lake, all dammed reservoirs. The Brazos is noted for high turbidity during times of high
discharge, and, conversely, high suspended chlorophyll concentration and high rates of water column primary
productivity at low flow (Roach et al., 2014).

From January 2012 through July 2013, weekly temperature, pH measurements, and water samples were

collected from the Brazos River at the Highway 60 bridge between Brazos and Burleson counties (VanPlantinga et
al., 2017). Water samples were measured for $\delta^{18}O$ and $\delta D$ using a Picarro L2120i cavity ringdown spectrometer at
the Stable Isotope Geoscience Facility at Texas A&M University. Calibration procedures, water $\delta^{18}O$, and
temperature values are given in VanPlantinga et al. (2017). Discharge data for the Brazos River at Highway 21 near
College Station (USGS 08108700) were obtained online from http://waterdata.usgs.gov/tx.

**2.2 Shell samples and analyses**

On August 9, 2013, four specimens each of *Amblema plicata* and *Cyrtonaias tampicoensis* were collected

live from the Brazos River near the Highway 60 bridge, from the sandy river bed shallower than 2 m depth. Mussels
were frozen, then shucked. Their shells were scrubbed, sonicated in water, and dried.



One specimen each of modern young adult *A. plicata* (labelled 3R5) and *C. tampicoensis* (TP3) were
randomly selected and analyzed. Based on stable and clumped isotope analyses, the shells are estimated to be 3-4
years old (VanPlantinga and Grossman, 2018). Specimens were sectioned, broken in two, and epoxied to glass
slides. Shell powder samples were collected with a New Wave Micromill using a 0.5 mm drill bit following the
methods of Dettman and Lohmann (1995). Two transects were sampled in each shell: one across the ventral
margin region (or VM, also referred to as the outer nacreous layer or ONL), and one across the INL region
(inner nacreous layer) as shown in Figure 2. Sample intervals were between 60 and 140 µm, with generally
shorter spacing for INL than ONL. About 60 µg per sample were reacted in a Kiel IV carbonate instrument
with phosphoric acid (specific gravity ≈ 1.925 g/cm$^3$) and the $CO_2$ analyzed on a Thermo Finnigan MAT253 mass
spectrometer in the Stable Isotope Geosciences Facility at Texas A&M University. Average analytical precision was
0.05‰ for $\delta^{18}O$ and 0.03‰ for $\delta^{13}C$.
For ICP-MS analysis, 20-160 µg of powder were dissolved in 2 mL of 2% nitric acid solution. ICP-MS was
performed on a Thermo Scientific high resolution inductively-coupled plasma mass spectrometer (HR-ICP-MS) at
Texas A&M University's Williams Radiogenic Isotope Geosciences Laboratory for the following nuclides: $^{25}Mg$,
$^{43}Ca$, $^{55}Mn$, $^{88}Sr$, $^{137}Ba$, and $^{56}Fe$. The USGS MACS3 coral reference standard was used as a validation standard (N =
12), and error analysis is provided in Table 1. An indium spike was added to all samples and standards to monitor
instrumental drift. Because the water samples were not filtered and were acidified for analysis after months in
storage, Mn concentrations may be underestimated. Below, we discuss the shell Mn/Ca values without relying
heavily on the water measurements.
Cross sections of TP3 and 3R5 shells were photographed with cathodoluminescence microscopy (CL)
using a Technosyn 8200 MKII cold cathode luminoscope following the methods of Roark et al. (2016). Samples
were exposed to a 400 nA and 20 kV beam with photograph exposure of about 30s. Photomosaics of the CL images
were arranged over high resolution scans of the shell cross sections and then analyzed with ImageJ software.
Brightness profiles were plotted from the same locations in the shells as the micro-drilled transects. Although some
CL photographs had shadows in the bottom left corners, shadows were cropped out in the INL regions. In order to
avoid shadows in the VM regions, data points in the shadows were identified on the plot in Figure 3A
(corresponding to the labeled regions in Figure 2) and removed from the CL data set analyzed in the cross-




correlation matrix (Table 2). The CL comparisons in Table 2 excluded 8 points from 3R5 and 1 point from TP3 from
the shadowy regions of the CL photomosaics. Normalized image brightness profiles were then compared with ICP-
MS results using Pearson's *r* values. To avoid false positives with multiple comparisons, we use a Bonferroni
correction for the overall level of significance α (0.05), divided by 52 comparisons, resulting in significance
threshold of $p < 10^{-3}$.

The distribution coefficient $D_{Me}$ represents the Me/Ca in the shell relative to the water Me/Ca where $D_{Me} =$

(shell Me/Ca) / (water Me/Ca). Ranges of shell $D_{Mg}$, $D_{Mn}$, $D_{Ba}$, and $D_{Sr}$ values were calculated using the minimum
and maximum shell Me/Ca values relative to the mean water Me/Ca values for water samples taken from April to
August of 2013 to overlap with the growth period of the shell VM trace element data.

**3 Results and discussion**
**3.1 Oxygen isotopes**

Stable isotope growth chronologies for specimens 3R5 and TP3 are shown in Figure 3 and explained in

detail in VanPlantinga and Grossman (2018). To develop these chronologies, we measured water temperature (T)
and $\delta^{18}O_{water}$ to predict shell $\delta^{18}O$ according to equations 1, 2, and 3 (Dettman et al.,1999, based on Grossman and
Ku, 1986).

$$1000 \ln (\alpha\ ^{aragonite}_{water}) = 2.559 \ \mathbf{x} \ (10^6 \ \mathbf{x} \ T^{-2}) + 0.715 \tag{1}$$

$$\alpha\ ^{aragonite}_{water} = \frac{(1000+\delta^{18}O_{aragonite_{VPDB}})}{(1000+\delta^{18}O_{water_{VSMOW}})} \tag{2}$$

$$\alpha\ ^{VSMOW}_{VPDB} = 1.0309 \ (\text{Gonfiantini et al., 1995}). \tag{3}$$

Because winter hiatuses and erratic summer growth patterns result in chaotic shell $\delta^{18}O$ patterns that complicate
$\delta^{18}O$ sclerochronology, we used clumped isotope thermometry to supplement $\delta^{18}O$ data (VanPlantinga and
Grossman, 2018).

Based on our shell chronology, the time interval represented by the trace element analyses is April to

August 2013. During this interval water temperature and $\delta^{18}O_{water}$ ranged from 13 to 34°C and -2.7 to 1.3‰,
respectively.  Daily average river discharge at the study site was 173-2230 cfs (cubic feet per second; USGS gage



08108700; https://waterdata.usgs.gov). The higher $\delta^{18}O_{water}$ values reflect increased summer evaporation combined
with increased proportion of flow from evaporated $^{18}O$-enriched Lake Whitney water, whereas lower values (as low
as -8‰) are the result of $^{18}O$-depleted precipitation and runoff (Chowdhury et al., 2010; VanPlantinga et al., 2017).

**3.2 Water chemistry**

Mean water Me/Ca values are presented in Table 1. Water dissolved ion concentration and electrical
conductivity results are shown in Figure 4A. The Sr, Ca, and Ba results track with the electrical conductivity
because Brazos River salinity is strongly controlled by the proportion of river flow discharged from Lake Whitney
(Chowdhury et al., 2010; VanPlantinga et al., 2017). Mg, Sr, and Ba correlated positively with Ca concentrations
and Mn correlated negatively with Ca (Rsq > 0.55, $p$ < 0.0007). Water Mn/Ca, Ba/Ca, and Sr/Ca values (mmol/mol)
significantly correlate with each other ($p$ < 0.00011), and further, Mg/Ca weakly correlates with Sr/Ca and Ba/Ca ($p$
< 0.015). USGS data for the Brazos River gage at Bryan, Texas (08108700) generally display an inverse relationship
between dissolved oxygen and discharge. On a linear scale, the element with the highest concentration, calcium,
showed the greatest variation (19-83ppm), but on a log scale magnesium concentration showed the most variation
(12ppb-20ppm; Figure 4). While the low water manganese concentrations (0.1-0.6 ppb) are consistent with Keeney-
Kennicutt and Presley's (1986) measurements (0.1-2.3 ppb), we will not draw conclusions based on the water Mn
data because our water samples were not filtered and acidified immediately upon collection. Turekian and Scott
(1967) attribute the suspended particulate manganese concentration in the Brazos River (690 ppm) to soil erosion, as
found in other river Mn studies (e.g., Shiller, 2002; Risk et al., 2010). The highest water Mn concentration values in
our study were from samples taken during times of high flow.

**3.3 Shell chemistry**

Table 2 explores relationships between environment, growth, and shell chemistry using Pearson's $r$ values.
Me/Ca values and distribution coefficients ($D_{Me}$) can differ between specimens 3R5 and TP3, and between the
ventral margin (VM) and inner nacreous layer (INL) of the same shell, especially with regard to Mg/Ca and Mn/Ca
(Table 1). Nevertheless, taken as a whole, the ranges in values are generally similar to those recorded in previous
studies of freshwater mussels (Carroll and Romanek, 2008; Geeza et al., 2018 and references cited therein), except
for Mg/Ca (Table 3). In addition, log of shell $D_{metal}$ values overlap with the results in Bolotov et al. (2015) for
metal/calcium partitioning in *Margaritifera*, except that their Mg/Ca values are 1-4 orders of magnitude lower than
ours (0.001-0.138).





Mg/Ca does not show any systematic patterns in our water data (Figure 3A), nor are there any systematic
variations in the Mg/Ca values of the shells, with erratic fluctuations over several orders of magnitude over the time
period studied (Figures 4B). Furthermore, taxonomic differences can be important.  For example, Mg/Ca values of
3R5 are about three times greater than those of TP3.
Previous studies of Mg/Ca and Sr/Ca indicate that shell trace elements may be heterogeneously distributed
in the shell mineral lattice and organic matrix depending on ontogenetic age, ultrastructure, and crystal fabric
(Schöne et al., 2011; Schöne et al., 2013). Brazos River water Mg/Ca is about half that in the Scioto River in Ohio
(Geeza et al., 2018), but our average shell Mg/Ca values are nearly an order of magnitude higher, resulting in
significantly higher $D_{Mg}$ estimates, than in the Ohio *Lamsilis cardium* shells. Differences in species or climate may
account for the variation in freshwater mussel $D_{Mg}$ values.
Sr/Ca correlates significantly with Mn/Ca in both shells. If Bonferroni corrections are not used as in other
studies (e.g., Vander Putten et al., 2000; Gentry et al., 2008; Izumida et al., 2011; Geeza et al., 2017), all but one Sr/Ca
relationship in Table 2 (with growth rate in 3R5) may be significant ($p < 0.05$), corroborating the common observation
that Sr/Ca correlates positively with temperature in aragonitic mollusk shells (e.g., Gillikin et al., 2005; Carré et al.,
2006; Sosdian et al., 2006). The Sr/Ca-temperature relationship was observed in lacustrine mussels by Izumida et al.
(2011), but was not observed in freshwater mussels from Ohio (Geeza et al., 2018) where there was significant shell-
water Sr/Ca relationship. The $D_{Sr}$ values from the shell ventral margin regions (0.08-0.19) overlap with $D_{Sr}$ values
reported in several previous studies (Carroll and Romanek, 2008; Bolotov et al., 2015; Geeza et al., 2017) as shown
in Table 3.
In terms of variation within and between shells, Sr/Ca is only slightly more concentrated in the INL than the
VM in both specimens. Figure 4 illustrates the similar patterns between Mn/Ca, CL brightness, shell growth rate,
Sr/Ca, and $\delta^{13}$C. There is a robust relationship between Sr/Ca and Mn/Ca in both the TP3 and 3R5 ventral margins
(Figure 3E). Sr/Ca values are similar between the two specimens, (Figure 4, Table 1).
As shown in Table 3 shell Ba concentrations in the ventral margin (45-2748 mg/kg) overlap with the range
reported in past studies (Carrol and Romanek, 2008; Bolotov et al., 2015; Geeza et al., 2017). Brazos shell $D_{Ba}$ values
(0.06-0.47) overlap with values given in other studies of freshwater mussels (Izumida et al., 2011; Bolotov et al., 2015;
Geeza et al., 2017). Out of the four Me/Ca parameters, Ba/Ca showed the second lowest mean values in the water and





in the shells. Ba/Ca values overlap with the range reported in past studies (Table 1; Carrol and Romanek, 2008;
Bolotov et al., 2015; Geeza et al., 2017 ). Ba/Ca are 29% higher in the Tampico specimen (TP3) than the threeridge
specimen (3R5). Ba/Ca was higher in the Tampico VM region than in the INL, but higher in the threeridge INL than
the VM.

While water Ba concentration is likely driven by the proportion of flow from Lake Whitney discharge, as

with Sr, Mg, and Ca (Chowdhury et al., 2010; VanPlantinga et al., 2017), the shell Ba/Ca values do not show any
systematic patterns. Previous authors have linked shell Ba/Ca to diatom productivity patterns (Vander Putten et al.,
2000; Lazareth et al. 2003). In the absence of periodic diatom blooms, Izumida et al. (2011) attributed their lacustrine
mussel shell Ba/Ca to growth rate. Our data do not point to a clear physical or physiological explanation for shell
Ba/Ca patterns in the Brazos River specimens.

Shell Mn/Ca values (mmol/mol) are shown in Figure 4A. Shell Mn concentrations (67-2308 mg/kg) overlap

with ranges reported in several studies (Nyström et al., 1996; Mutvei and Westermark, 2001; Markich et al., 2002;
Verdegaal, 2002; Ravera et al., 2003; Langlet et al., 2007; Carroll and Romanek, 2008; Bolotov et al., 2015; Zhao et
al., 2017a; Geeza et al., 2017). The $D_{Mn}$ values from the shell ventral margin regions in this study (13-84) overlap with
$D_{Mn}$ ranges reported in Geeza et al. (2018) and Bolotov et al. (2015) but are much higher than other studies where $D_{Mn}$
< 1 (Markich et al., 2002; Verdegaal, 2002; Carroll and Romanek, 2008). The average $D_{Mn}$ values of the 3R5 and TP3
INL regions are higher (~80-200). Compared to thermodynamic predictions for abiogenic aragonite, biogenic
aragonite has relatively high substitution rates of $Mn^{2+}$ for $Ca^{2+}$ in the mineral lattice (Soldati et al., 2016). Relatively
high $D_{Mn}$ values (>10) in biogenic aragonite, as reported here, suggest a physiological process of concentrating $Mn^{2+}$
during biomineralization. The influence that factors such as species differences, environment, and ontogeny have on
$D_{Mn}$ remain to be determined.

Mn/Ca is significantly higher and more variable in the INL than VM (or ONL) regions in both TP3 and 3R5

specimens (Table 1). Figure 4B shows shell INL Mn/Ca and water Mn/Ca for 2012-2013. Siegele et al. (2001)
suggested that shell growth rings have elevated manganese and organic matter content in *Hyridella depressa*, and they
inferred different shell chemistry and mineralization processes between the shell umbo and ventral margin. Carroll
and Romanek (2008) suggest that differences between INL and ONL trace element values may come from higher
rates of dissolution and reprecipitation in the INL than in the ONL. Oeschger (1990) suggested that anaerobiosis





contributes to the internal dissolution of the shell in *Arctica islandica*. Some biomineralization models indicate that
the INL is exposed to extrapallial fluid of a different chemical composition than the EPF in contact with the shell ONL
region (Schöne and Krause, 2016). If this is the case, then the shell INL trace element values may be less appropriate
for environmental reconstruction than the ONL region. Higher Mn/Ca in the INL than in the VM regions of the Brazos
River specimens indicates physiological control on the distribution of Mn in the shell. Mn/Ca values are on average
27% higher in the Tampico specimen (TP3) than the threeridge specimen (3R5). This may reflect species or individual
differences in metabolic rate.
Shell Mn/Ca correlates inversely with log of river discharge (Table 2), allowing for the reconstruction of
times of high and low flow. Figure 3F reconstructs trends in log of Brazos River discharge ($\log_{10}Q$) from Mn/Ca in
TP3 ($\log_{10}Q = -1.11 \times$ Mn/Ca$_{shell}$ + 3.17) and in 3R5 ($\log_{10}Q = -1.22 \times$ Mn/Ca$_{shell}$ + 2.99). The reconstruction is more
accurate in the summer but overestimates observed discharge in the spring, possibly due to seasonal changes in
water Mn/Ca or biological controls on shell Mn/Ca. Because of 1) the higher Mn/Ca in the INL relative to the VM
regions in the shells, and 2) the strong relationship between shell Mn/Ca and river discharge, we infer both physical
and biological controls on shell Mn/Ca, as discussed below.
Previous studies have used shells chemistry to reconstruct river discharge such as by linking high runoff
events to elevated suspended Mn from soil erosion (Risk et al., 2010). Many sclerochronological reconstructions of
discharge are based on stable oxygen isotopes (Mueller-Lupp et al., 2003; Dettman et al., 2004; Versteegh et al., 2011;
Ricken et al., 2003; Kelemen et al., 2018). Our study indicates that Brazos River mussel activity patterns (feeding
and/or metabolic rate) are influenced by discharge rates and that these variations are recorded in the trace element
composition, particularly Mn/Ca, of the shell mineral. Here we reconstruct river discharge variation and distinguish
times of low and high flow using shell Mn/Ca values (Figure 3F).
**3.4 Cathodoluminescence**
Cathodoluminescence (CL) is a common tool for mapping the distribution of manganese in biogenic
carbonates (Barbin, 2000). Lattice-bound Mn caused greenish-yellow luminescence under CL. The CL images
reveal alternating bright green-yellow and dim banding that generally correlates with the pattern of light and dark
banding in plane light (Figure 2). The results verify that the Mn is lattice-bound (Table 2) and reveal the complex



cyclicity of Mn distribution in the shell (Lartaud et al., 2009). CL brightness also weakly correlates with Sr/Ca and
G (growth rate) in both shells.

**3.5 Dissolved and particulate sources of Mn**

Manganese incorporated into the mussel shells may be derived from dissolved Mn or ingested particulate

Mn. Several factors affect manganese concentration and flux in the environment. Reducing conditions, low DO, and
low pH increase manganese solubility (Tebo et al., 2004). Microbial activity combined with high nutrient flux and
low rates of water column mixing can cause hypoxia, reducing conditions, and elevated dissolved $Mn^{2+}$
concentration (Zhao et al., 2017a). Other factors influencing Mn availability include photo-inhibition of $Mn^{2+}$-
oxidizing bacteria, reductive dissolution from sunlight (Sunda and Huntsman, 1994), primary production, benthic
decomposition, algal uptake of dissolved $Mn^{2+}$ (Sunda and Huntsman, 1985), and influx of allochthonous dissolved
$Mn^{2+}$ (Langlet et al., 2007).

Shell manganese could be influenced by point sources such Lake Whitney or the Little River. Lake

Whitney and Little River manganese concentrations are near the mean values of the Brazos River (~0.2ppb; this
study). Lake Whitney has periodic brown algae blooms (Roelke et al., 2011). However, if Lake Whitney was the
driver of shell Mn/Ca patterns, then the water Mn/Ca patterns would not be inversely related to water Sr/Ca, Ba/Ca,
and Mg/Ca. Elevated $\delta^{13}C$ in the shells during the summer of 2013 was interpreted as an indication of heightened
Lake Whitney influence on river flow and chemistry during drought conditions (VanPlantinga and Grossman, 2018;
VanPlantinga et al., 2017). There is a correlation between $\delta^{13}C$ and Mn/Ca in 3R5 but not in TP3. There is not yet
sufficient evidence to indicate that Lake Whitney or the Little River are point sources of shell manganese, nor to
explain the striking inverse shell Mn/Ca - river discharge relationship, but the point source hypothesis cannot be
ruled out given the important role Lake Whitney plays in downstream river chemistry (VanPlantinga et al., 2017).

Dissolved $Mn^{2+}$ is the most bioavailable form of manganese (Campbell, 1995). Shell Mn/Ca values have

been attributed to variations in dissolved $Mn^{2+}$ in the water column (Frietas et al., 2006; Barats et al., 2008) and in
the sediment porewater (Zhao et al., 2017a). As mentioned earlier, experimental studies have confirmed that
dissolved $Mn^{2+}$ content is recorded in shell Mn/Ca (Jeffree et al., 1995; Hawkes et al., 1996; Markich et al., 2002;
Langlet et al., 2006; Lartaud et al., 2010). However, the low dissolved oxygen conditions in the Brazos River, which
should favor high dissolved $Mn^{2+}$, occur at times of high flow (USGS 08108700 gage data) when shell Mn/Ca is





relatively low.  Redox conditions in the water column do not explain the shell Mn/Ca patterns, and we lack the data
to evaluate the hypothesis that sediment porewater drives shell Mn/Ca.
Particulate Mn, bound to organic or inorganic particles, can also be a source of Mn in shells. The inverse
relationship between water Ca and Mn concentrations indicates that Mn flux into the water may be related to runoff
from local rain storms, in contrast to the Ca sourced from the upstream reservoir Lake Whitney (Chowdhury et al.,
2010; VanPlantinga et al., 2017). Bilos et al. (1998) attributed elevated clam soft tissue Mn to higher turbidity and
ingestion of Mn-bearing inorganic particles. Because Mn/Ca is inversely correlated with log of discharge in this
study, inorganic particles (suspended during at times of high flow) are probably not the source of Brazos River
mussel shell Mn/Ca.
Previous studies have attributed bivalve shell Mn/Ca to ingestion of Mn-bearing organic particles such as
phytoplankton. Vander Putten et al. (2000) and Lazareth et al. (2003) found significant shell Ba/Ca-Mn/Ca
correlations in estuarine bivalves indicative of diatom ingestion. Brazos River phytoplankton are typically not
diatoms (Roelke, personal communication) and there is no shell Ba/Ca-Mn/Ca relationship in our data. Geeza et al.
(2018) examined oxygen, chlorophyll, and pH as a proxy for primary productivity (based on Goodwin et al., 2018),
but did not find correlations with shell Mn/Ca. Nevertheless, they could not rule out a phytoplankton or microbial
manganese reduction (Lovley and Phillips, 1988) influences on their shell Mn/Ca values.
Roach et al. (2014) found elevated chlorophyll concentrations in the Brazos River near our study site at
times of low discharge in 2010-2012, with suspended chlorophyll concentration significantly higher than benthic
chlorophyll (40-50 mg/L compared to ~11 mg/L), and about 5-10 times higher than the other rivers in their study.
Roach (2013) attributed river chlorophyll abundance to lengthened residence time, emphasizing that sediment
scouring and turbidity from high discharge limit phytoplankton growth (Wissmar et al. 1981; Steinman and
McIntire, 1990). River mussels have been observed to preferentially inhabit refugia with low rates of shear stress
(Layzer and Madison, 1995; Strayer, 1999; Howard and Cuffey, 2003). This may correspond to elevated manganese
concentrations in sediment porewater as in Zhao et al. (2017a). However, little is known about the spatial and
temporal variation and chemical composition of Brazos River phytoplankton, suspended load, and colloids in the
flowing river water and the sediment porewater. Future work should characterize these variables.
**3.6 Manganese accumulation in shells**

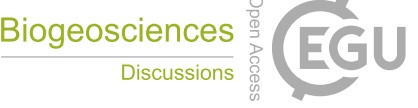

Shell Mn/Ca is potentially determined by a combination of environmental chemistry (e.g., water and
particle chemistry), physical conditions (e.g., temperature and discharge), and the behavior of the organism (e.g.,
feeding rate, growth rate, and reproductive investment). Zhao et al. (2017b) offer a similar interpretation of
*Corbicula fluminea* shell Ba/Ca based on laboratory experiments. In terms of feeding behavior, mussels selectively
ingest organic matter during filter feeding (Hawkins et al., 1996). Zhao et al. (2017a) propose that manganese
bioaccumulation in lacustrine mussels is enhanced by deposit feeding (Vaughn and Hakenkamp, 2001; Cahoon and
Owen, 1996). The elevated concentrations of suspended chlorophyll relative to benthic chlorophyll at our study
location at times of low flow are conditions favorable for suspension feeding (Roach et al., 2014). The propensity in
river mussels to inhabit refugia with minimal shear stress (Layzer and Madison, 1995; Strayer, 1999; Howard and
Cuffey, 2003) supports the hypothesis that Brazos River mussels thrive under conditions of low discharge with high
concentrations of suspended particulate organic matter to feed on.
It is important to consider the physiological processes and soft tissues potentially involved in manganese
bioaccumulation. Langlet et al. (2007) suggest that soft tissues concentrate Mn derived from the digestion and
absorption of organic particles and this may lead to elevated Mn/Ca values in the shells. Acidic pH in the gut makes
ingested particulate Mn bioavailable so that it can then accumulate in mollusk soft tissue and the shell (Arifin and
Bendell-Young, 2000; Owen, 1996). Nott and Nicolaidou (1993) found that a substantial 67% of ingested manganese
is not recovered in feces of the mollusk *Nussarius rericulatus*, and therefore it is absorbed through the digestive tract.
Mollusk bioaccumulation of heavy metals through the gills and digestive glands is well documented and supports the
hypothesis that shell manganese can bioaccumulate via food ingestion (Domouhtsidou et al., 2000; Dimitriadis et al.,
2003; Einsporn and Koehler, 2008).
The shell Sr/Ca-Mn/Ca may indicate a relationship between metabolic rate, inferred from Sr/Ca, and feeding
rate, inferred from Mn/Ca. Metabolic rate is influenced by factors such as ontogeny, reproductive investment,
environmental stress (drought, flood, predation), and seasonal feeding patterns (Bayne et al., 1989). Brazos mussel
shell Sr/Ca may reflect metabolic patterns that cause varying rates of ion transport into the EPF as hypothesized in
Carré et al. (2006). Zhao et al. (2016) experimentally changed dissolved $Ca^{2+}$ concentrations and used lanthanum and
Verapamil to artificially inhibit $Ca^{2+}$ channels in the freshwater bivalve *Corbicula fluminea* and concluded that $Mn^{2+}$
and $Ca^{2+}$ compete to cross ion channels during biomineralization. In light of the important role ion channels play in



biomineralization, the Sr/Ca-Mn/Ca correlation in the Brazos River shells points to a relationship between metabolic
rate and feeding rate. However, the physiological mechanism of ion channels does not diminish the importance of
environmental factors such as water chemistry and redox conditions in determining shell Me/Ca values, as indicated
in many studies (Campbell, 1995; Jeffree et al., 1995; Hawkes et al., 1996; Markich et al., 2000; Frietas et al., 2006;
Langlet et al., 2006; Barats et al., 2008; Lartaud et al., 2010; Zhao et al., 2017a; and for Sr/Ca in the case of Geeza et
al., 2017).

Little is known about the pathway that environmental manganese takes from ingestion to shell mineralization.

Amorphous calcium carbonate (ACC), conveyed by hemocytes to the mantle, is the precursor to the shell mineral
(Addadi et al., 2003; Mount et al., 2004; Li et al., 2016). The ACC has higher Mn and other metal concentrations than
the shell mineral (Thomson et al., 1985; Jacob et al., 2008). Initial manganese exposure may be primarily to the gills,
hemolymph, mantle, or digestive tract, and it may travel to the site of biomineralization via particulate or dissolved
forms through the hemolymph and mantle tissue. Marin et al. (2012) describe intercellular and intracellular dissolution
and formation of ACC granules in the mantle tissue, potentially blurring the distinction between granule and calcium
ion channel transport of trace metals to be incorporated into the shell mineral lattice. Dissolved and ACC-bound $Mn^{2+}$
physiological pathways should be investigated further.
**5 Conclusions**

Mn/Ca values for Brazos River mussel shells showed a cyclical pattern revealed by time series analyses

and cathodoluminescence, which maps a pattern similar to the growth bands. Mn/Ca correlated inversely with
discharge, allowing for a reconstruction of river discharge patterns during the study period. Mn/Ca is likely
influenced by ingestion rates of Mn-bearing suspended particulate organic matter because shell Mn/Ca is high when
river discharge and turbidity are low, ruling out inorganic particles as the control on shell Mn/Ca. The shell Mn-Sr
relationship and the evidence of high suspended chlorophyll at times of low flow (Roach et al., 2014) point to
elevated metabolic activity and likely increased feeding rate in response to food abundance, and possibly lower
shear stress and turbidity, at times of low flow. Future research on shell and water chemistry should: 1) further the
scientific understanding of river plankton, suspended colloidal and sediment porewater manganese variation; 2)
resolve taxonomic $D_{Mn}$ differences; and 3) elucidate specifically why different mussels in different environments
have $D_{Mn}$ values <1 and others $D_{Mn}$ values are >10.


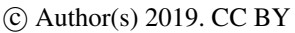



**Code/Data Availability**

Data are available on earthchem.org.

**Authors' Contributions**

A. VanPlantinga collected and analyzed data, made plots and tables, and wrote and revised the manuscript. E.
Grossman provided funding and edited the manuscript, the plots, and the tables.

**Competing Interests**

The authors have no competing interests to declare.

**Acknowledgments**

The authors would like to thank the Michel T. Halbouty Chair in Geology at Texas A&M University for
supporting this research. Luz Romero, Franco Marcantonio, and the Texas A&M AgriLife Extension Soil, Water
and Forage Testing Laboratory analyzed shell and/or water samples. Charles Randklev and Eric Tsakiris collected
the mussel specimens. Ann Molineux from The University of Texas Jackson School Museum of Earth History
provided historical specimens from the Singley-Askew Collection. Charles Randklev and Robert G. Howells
provided helpful perspective on mussel ecology. Ben Passey, Naomi Levin, Huanting Hu, Haoyuan Ji, Sophie
Lehmann, Dana Brenner, and Lai Ming provided valuable assistance at the Johns Hopkins University Stable Isotope
Lab. Chris Maupin, Lauren Graniero, Andrew Roark, and Brendan Roark helped run isotope samples at the Stable
Isotope Geoscience Facility at Texas A&M University. Clumped isotope analyses were financed by NSF grant
EAR-1226918. Data reported here are in the following online repository: earthchem.org. We thank reviewers
Christopher Romanek and David Dettman for their helpful comments.

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



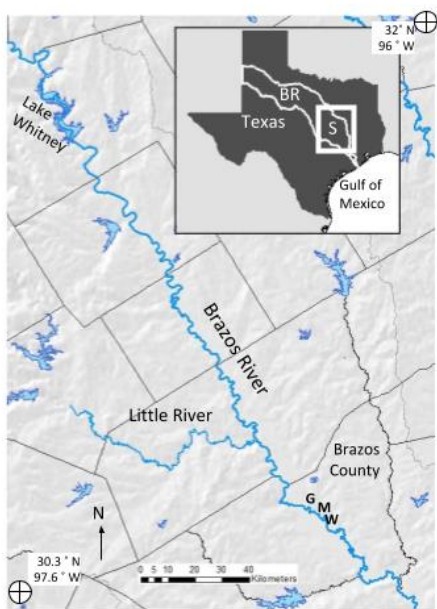

**Figure 1.** Study area. Inset: Map of Texas, Brazos River watershed (BR), and study area (S). The map reaches from Lake Whitney in the north to Brazos County in the south, showing the water collection (W), mussel collection (M), and gage (G, USGS gage 08108700) locations.




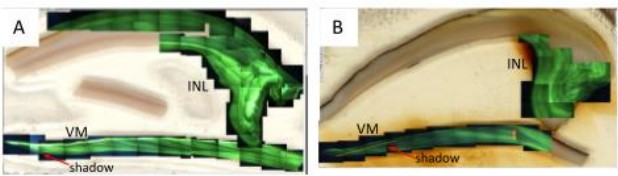

**Figure 2.** Cathodoluminescence (CL) photomosaics for TP3 (A), 3R5 (B). Thin yellow lines in A and B are the transects analyzed with ImageJ. The sampled INL (inner nacreous layer) and VM (ventral margin) regions are labeled in A and B.



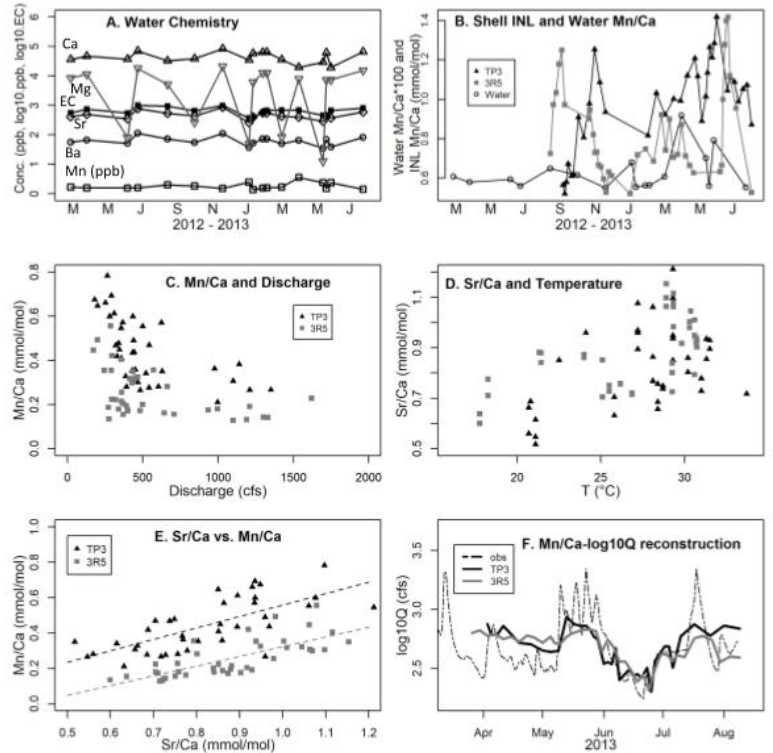

**Figure 3.** (A) Water chemistry measurements from the Brazos River (2012-2013); empty squares =
Mn (ppb), empty circles = Ba (log10 of ppb), empty upright triangles = Ca (log10 of ppb), filled
squares = electrical conductivity (log10 of μS), inverted empty triangles = Mg (log10 of ppb),
diamonds = Sr (log10 of ppb). (B) Water Mn/Ca (100*mmol/mol) and shell INL Mn/Ca in
mmol/mol. (C) Discharge vs. Mn/Ca. (D) Temperature vs. Sr/Ca. (E) Shell Sr/Ca vs. shell Mn/Ca
values. (F) Log10 of river discharge (Q) and reconstructions of log10 (Q) based on the shell
Mn/Ca-Q relationship.




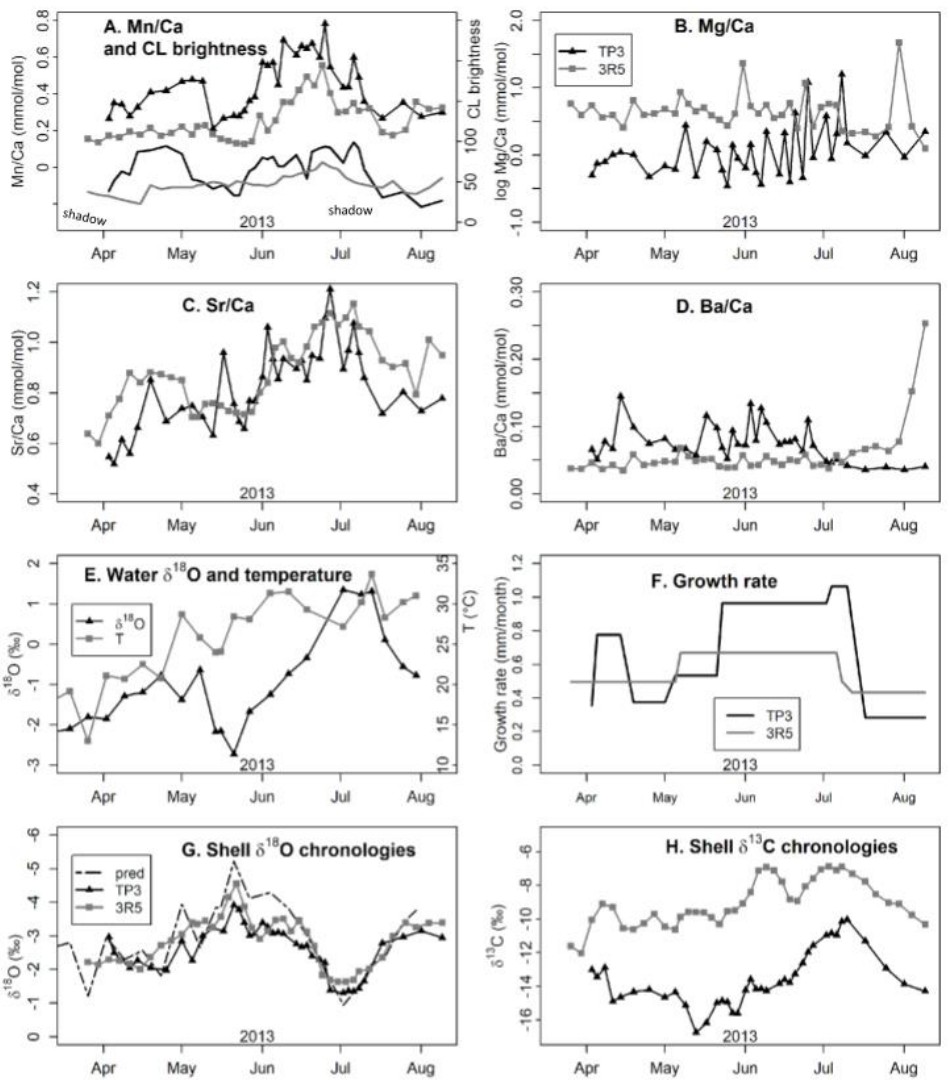

**Figure 4.** TP3 (black triangles and/or black line), and 3R5 (gray squares and/or gray line) values for shell Mn/Ca and CL (A); shell Mg/Ca (B); shell Sr/Ca (C); shell Ba/Ca (D); water δ18O and temperature (E); estimated shell growth rate (F); shell δ18O chronologies for TP3, 3R5, and predicted aragonite δ18O (G); and shell δ13C chronologies (H). The shell isotope chronologies are described in detail in Van Plantinga and Grossman (2018).




**Table 1.** Summary of MACS3 check standard results and error analysis and Brazos River water and shell results by shell region for trace metal Me/Ca values and calculated partition coefficients D(Me/Ca of shell/water).

|  | Mn/Ca | Sr/Ca | Ba/Ca | Mg/Ca |
|---|---|---|---|---|
| **MACS3 check standard and uncertainty analysis** | | | | |
| Mean* | 1.07 | 8.70 | 0.05 | 7.70 |
| Std. dev.* | 0.082 | 0.187 | 0.004 | 0.151 |
| RSD | 7.6% | 2.2% | 7.6% | 2.0% |
| Precision | 2.2% | 0.6% | 2.2% | 0.6% |
| Accuracy | 3.5% | 0.7% | 6.5% | 3.9% |
| Cert. values* | 1.11 | 8.76 | 0.05 | 8.01 |
| *mmol/mol Ca | | | | |
| | | | | |
| **Mean Brazos River and mussel shell values (mmol/mol Ca)** | | | | |
| Water | 0.006 | 5.45 | 0.46 | 292.9 |
| TR5VM | 0.26 | 0.88 | 0.058 | 6.86 |
| TR5INL | 0.83 | 1.13 | 0.085 | 0.79 |
| TP3VM | 0.44 | 0.82 | 0.072 | 2.07 |
| TP3INL | 1.29 | 1.05 | 0.058 | 13.63 |
| | | | | |
| **Mean distribution coefficients** | | | | |
| TR5VM | 27 | 0.14 | 0.11 | 0.02 |
| TR5INL | 89 | 0.18 | 0.16 | 0.002 |
| TP3VM | 47 | 0.13 | 0.14 | 0.006 |
| TP3INL | 135 | 0.16 | 0.11 | 0.04 |






**Table 2.** $r^2$ and $p$ values for relationships between log10 of discharge (log Q), temperature (T), river water $\delta^{18}O_w$, growth rate (G in mm/month), $\delta^{18}O$, $\delta^{13}C$, Mn/Ca, Sr/Ca and CL for specimens TP3 and 3R5. $R^2$ and $p$ values are in **bold** if $p$ is less than the Bonferroni-corrected α value of 0.05 / 52 = 0.001. Gray italicized $p$ values exceed the Bonferroni-corrected α value.

| | CL $R^2$ | CL $p$ | Mn $R^2$ | Mn $p$ | Sr $R^2$ | Sr $p$ | G $R^2$ | G $p$ | $\delta^{18}O$ $R^2$ | $\delta^{18}O$ $p$ |
|---|---|---|---|---|---|---|---|---|---|---|
| **TP3** | | | | | | | | | | |
| log Q | 0.31 | **3.7E-04** | 0.49 | **1.6E-06** | 0.13 | 2.6E-02 | *0.02* | *4.2E-01* | 0.20 | 6.1E-03 |
| T | *0.00* | *7.4E-01* | 0.18 | 9.8E-03 | 0.26 | 1.2E-03 | *0.06* | *1.4E-01* | *0.04* | *2.1E-01* |
| $\delta^{18}O_w$ | 0.13 | 3.0E-02 | *0.07* | *1.0E-01* | 0.18 | 8.4E-03 | | | 0.56 | **1.2E-07** |
| G | 0.26 | 1.4E-03 | 0.27 | **9.5E-04** | 0.24 | 2.3E-03 | | | | |
| $\delta^{18}O$ | | | *0.06* | *1.6E-01* | 0.12 | 3.7E-02 | | | | |
| $\delta^{13}C$ | 0.14 | 2.5E-02 | *0.09* | *7.4E-02* | 0.20 | 5.9E-03 | *0.10* | *5.6E-02* | | |
| CL | | | 0.43 | **1.2E-05** | 0.34 | **1.7E-04** | | | | |
| Sr/Ca | | | 0.49 | **1.5E-06** | | | | | | |
| **3R5** | | | | | | | | | | |
| log Q | 0.16 | 2.1E-02 | 0.45 | **2.3E-05** | 0.29 | 1.6E-03 | *0.00* | *9.1E-01* | *0.12* | *5.6E-02* |
| T | 0.18 | 1.5E-02 | 0.27 | 2.3E-03 | 0.30 | 1.2E-03 | *0.00* | *7.4E-01* | *0.03* | *3.1E-01* |
| $\delta^{18}O_w$ | *0.02* | *4.6E-01* | 0.17 | 2.0E-02 | 0.53 | **2.0E-06** | | | 0.65 | **2.6E-08** |
| G | 0.21 | 8.4E-03 | *0.04* | *2.9E-01* | *0.01* | *6.4E-01* | | | | |
| $\delta^{18}O$ | | | 0.22 | 7.3E-03 | 0.58 | **4.9E-07** | | | | |
| $\delta^{13}C$ | 0.20 | 1.1E-02 | 0.25 | 3.2E-03 | 0.53 | **2.7E-06** | *0.06* | *1.8E-01* | | |
| CL | | | 0.61 | **1.6E-07** | 0.31 | 1.0E-03 | | | | |
| Sr/Ca | | | 0.56 | **7.6E-07** | | | | | | |




Table 3. Comparison of shell chemistry and shell/water distribution coefficient results ($D_{Me}$) with past studies (based on Geeza et al., 2017).

| Reference | Sr (mg/kg) | $D_{Sr}$ | Ba (mg/kg) | $D_{Ba}$ | Mg (mg/kg) | $D_{Mg}$ (×10⁻³) | Mn (mg/kg) | $D_{Mn}$ | Dissolved Mn |
|---|---|---|---|---|---|---|---|---|---|
| Faure et al. (1967) | | 0.22–0.28 | | | | | | | |
| Nyström et al. (1996) | 300–600 | | | | | | 10–600 | | |
| Mutvei and Westermark (2001) | | | | | | | 400–6000 | | |
| Markich et al. (2002) | | | | | | | 300–1700 | 0.6 | |
| Verdegaal (2002) | 120–220 | | 0.1 | | | | 100–700 | 0.5 | |
| Bailey and Lear (2006) | 700–1000 | 0.28 | | | | | | | |
| Langlet et al. (2007) | | | | | | | 100–1000 | | |
| Ravera et al. (2007) | | | | | | | 200–800 | | |
| Carroll and Romanek (2008) | 120–2000 | 0.17–0.26 | 60–400 | 0.05 | | | 80–1700 | 0.2–0.5 | 36–188 |
| Izumida et al. (2011) | | 0.18–0.22 | | 0.069–0.086 | 150–500 | 0.30–0.42 | | | |
| Bolotov et al. (2015) | 345–595 | 0.15–0.26 | 32–92 | 0.2–0.6 | 23–43 | 0.2–0.4 | 139–469 | 10–300 | |
| Zhao et al. (2017) | 1130–1380 | | | | | | 400–1800 | | 70–1400 |
| Geeza et al. (2017) | 820–3343 | 0.16–0.20 | 15–270 | 0.11–0.14 | 26–1200 | 0.3–0.8 | 120–1250 | 32–42 | 10–60 |
| This study | 430–5279 | 0.08–0.19 | 45–2748 | 0.06–0.47 | 36–89718 | 1–138 | 67–2308 | 13–84 | 0.1–0.6 |

| Water Data | Sr | | Ba | | Mg | | Mn | | |
|---|---|---|---|---|---|---|---|---|---|
| Water Conc. (ppb), this study | 255–852 | | 34–112 | | 12–20837 | | 0.1–0.6 | | |
| Water Me/Ca (mmol/mol) | 3–10 | | 0.2–1.0 | | 0.7–714.9 | | 0.001–0.022 | | |
