# Peer review of "Trace elements in mussel shells from the Brazos River, Texas: environmental and biological control"

_Biogeosciences, 2019_

## Referee Comment (RC1) · Anonymous Referee #1 · 16 Oct 2019

Review of the manuscript "Trace elements in mussel shells from the Brazor River, Texas..."

The paper addresses environmentally important question potentially suitable for Biogeosciences.

The title is not correct: this work is about Mn essentially, rather than trace elements Many references are incomplete

L10: does it simply mean that Sr correlated with Mn in shells?

Introduction: The novelty of this study and motivation behind this work are unclear.

[Figure]

Why river Mn flux is important at all? Indirect assessment of this flux via shells is not the easiest way...

Methods: How the samples were processes after dissolution; were they filtered? Was the dissolution complete?

Eqn 1-3: Unclear why this information is needed

Sampling (L149-150): The water samples were not filtered and acidified. As such, metal concentration (except probably Ca and Sr) in river water could not be measured and distribution coefficients do not make sense. Moreover, the whole main motivation of this study - reconstruction of Mn flux in the river - becomes compromised. As such, the distribution coefficients oven in Table 3 may not be usable.

In Fig 3C, use log scale for discharge. How good is this correlation? What about correlation with temperature?

L254-259: Another issue is what is Mn concentration in the lake hypolimnion? If the lake is seasonally stratified, then, during the overturn, the bottom Mn-rich waters can feed the river thus dramatically increasing the Mn concentration in the river water.

L257-258: The argument is unclear. In Fig. 3E, Mn/Ca is not inversely related to Sr/Ca. Please show the relationships.

L270-271: The low flow may provide enhanced $Mn2+$ input from the riparian and hyporheic zone

L277: notice here that the maximal suspended load is usually observed at high discharge

L288: Chl a of mg/L concentration is really high. May be a misprint here and the concentrations are in $\mu$g/L?

L293-295: As a conclusion to section 3.5, this is extremely discouraging. It looks like one cannot yet discuss the sources of Mn for shells, so this section is useless...

L312 Owen1996 is not in the ref list

L325-329: May be place this information in the Introduction. Again, this sentence is very discouraging: how is it related to particulate case analyzed in this work? What is more important, according to authors, in Brazes River: physiological mechanisms or environmental factors?

Rewrite L 347-348

L349-350: This is not sufficiently discussed and the whole story of DMn can be compromised by inadequate sampling

---

## Referee Comment (RC2) · Anonymous Referee #2 · 18 Oct 2019

Alexander A. VanPlantinga & Ethan L. Grossman prepared an interesting work which aimed to provide a better understanding of how environmental and biological factors affect the trace elemental composition in freshwater mussel shells. Yet, my feeling is that the authors may rephrase the title as proposed before, for example, focusing on the application of freshwater mussel shell trace elements to retrospective monitoring of riverine discharge. If the structure of the work went this way, the authors may likely shy away of one of the major flaws of this work – water samples were not filtered and acidified thereby hindering the reliable calculation of distribution coefficient. Actually, I think this parameter is not closely related to the story, as there are already several studies which reported such data even in freshwater bivalve species (Mg, Sr, Mn as

well as Ba). As such, I would urge the authors to consider these suggestions, especially given that these results are very interesting.

Several minor comments listed below: Introduction – should be rephrased to give a clear clue of the work. Line 17, The definition of sclerochronology is not complete, simply referring to Wikipedia and references therein. Line 56, why did the authors expect a relationship between Sr/Ca and temperature?

Method Lines 99-101, I appreciate that authors are not trying to hide this flaw in the manuscript but this indeed strongly limits the strength of conclusions.

Results & Discussion Lines 142-143, if the authors reported and discussed trace elemental concentrations in shells in the form of element-to-calcium ratio (Me/Ca), so please stay consistent throughout the manuscript. In fact, calcium concentration in bivalve shells can most likely be consistent over time. Line 173, in the leading sentence, if you are talking about "significantly", "p" value should always be given. Section 3.6, in my opinion, the authors should devote efforts to discussing the potential of Mn/Ca as an indicator of riverine discharge.

Conclusion The leading two sentences are nicely written to give a clear summary of the work. In line 345, removing Roach et al. (2014).

---

## Referee Comment (RC3) · Chris Romanek (Referee) · 21 Oct 2019

Review of bg-2019-323: Trace elements in mussel shells from the Brazos River, Texas: environmental and biological control by VanPlantinga and Grossman

This manuscript describes an investigation to better understand the trace element geochemistry of two freshwater mussel shells collected from the Brazos River TX in 2013. The investigators serially sampled the inner nacreous layer (INL) and outer ventral margin (VM) of a shell of specimen of Amblema plicata and Cyrtonaias tampicoensis and they analyzed the carbonate for Mg, Ca, Sr, Ba and Mn concentration. The data were placed within a temporal framework using previous oxygen isotope data (VanPlantinga

and Grossman, 2018) and compared to other environmental data (e.g., temperature, water chemistry, water discharge rate) to better understand the origin(s) of the geochemical signatures locked in the shell. The principal outcome of the study focused on shell Mn/Ca records which correlated inversely with river discharge allowing for a reconstruction of river discharge patterns. The investigators conclude that Mn was attained primarily through the ingestion of Mn-bearing particulate organic matter.

This manuscript addresses subject matter that is of general interest to the Biogeoscience community and it identifies a novel proxy of river discharge in the geochemical record of the shell. The investigators thoughtfully consider and evaluate the relevant peer reviewed literature within the context of their study.

The manuscript is relatively well written, although I had some trouble with the PDF because the figures contained a lot of information that could not be seen easily on a paper copy.

The amount of information provided on shell geochemistry and the various correlations and comparisons made me feel overwhelmed at times. The Mg, Sr, and Ba records (as well as ïĄď13Cand CL) appear ancillary at times and it is unclear if they provide meaningful supporting evidence for the major conclusions of the manuscript.

I recommend that the investigators reconsider what is really needed to support the main conclusion(s) of their study. This should help to sharpen the take home message of the study. A more highly focused revision that centers on the Mn records would be valuable to the scientific community and make a meaningful contribution to the published literature. Data that may be excluded from a revision could likely form the basis for separate manuscript(s).

Other comments, identified by line number (L), are provided below: General comment: Each time a geochemical profile is presented or discussed the investigators should identify whether it is from the inner nacreous layer or the ventral margin. Sometimes this distinction is clearly made while at other times it appears ambiguous.

[Figure]

L17: The first sentence that defines the word "sclerochronology" is inaccurate. Sclerochronology is more specific than "... the study of the physical and chemical properties of invertebrate hard parts...", it involves the temporal context in which these properties are considered.

L59: Methods: The physicochemical water sampling procedures need to be explained in greater detail, or cited properly. This extends to L99-101 where the statement is made that "... water samples were not filtered and acidified for analysis after months in storage." The implications of this unusual sampling strategy should be explained in greater detail. How severely is the Mn data compromised? How are other elements affected in addition to Mn?

L74: Hydrogen isotope compositions are not discussed in the text so a description of sample/analysis procedures in not warranted. Also, anytime delta notation is used, e.g., ïĄď18O, the word "value" should follow it.

L90 and L94: It appears the sampling resolution for stable isotopes (60 ïĄ■g) and ICP-MS analysis (20-160 ïĄ■g) differed. The implications of this should be addressed somewhere in the text.

L102: Were the CL images taken before or after the sampling of the shell for isotopic and elemental analysis? The observation of "shadows" should be explained a little better. This could be done in a better description of the CL imaging in general. What is the ultimate purpose of CL imaging? It appears the investigators wished to correlate brightness to measured Mn concentration in the shell. Was the sampling resolution for ICP-MS comparable to the width of bright and dark bands in the CL images?

L121: Oxygen isotopes. The first paragraph can be reduced if the manuscript needs to be shortened.

L146-148: Variance in a data set is independent of the scale over which the data are considered, e.g., the Mg data set (12 ppb – 20 ppm) is more variable than the Ca data

set (19-83 ppm) regardless of whether the data are considered on a linear or log scale. Also, reference to Fig. 4 (L148) is for Me/Ca ratios and not absolute concentrations.

L155: The word "shell" should proceed the word "growth".

L162: DMg values reported in Table 3 range from 1-138 (10-3) or 0.001-0.0138, not "0.001-0.138" as stated in the text.

L163: "Mg/Ca does not show any [temporal] systematic trend in our water data. . ."

L190: Sentence starting with "Ba/Ca values. . ." is redundant given L186 . L211: Variability in shell Mn/Ca in not reported in Table 1.

L212: I think the reference should be to "Figure 3B" here and not "Figure 4B".

L237: The final statement in the paragraph is introductory and should come earlier in the manuscript.

L238: The section entitled Cathodoluminescence is too brief. It should be expanded and integrated better in the text.

L260: Is this the only place where the carbon isotope data are discussed except for L184-185? The carbon isotope data should be integrated more fully in the text, e.g., why are there correlations between shell growth and Sr/Ca vs ïĄď13C values?

L283: Goodwin et al. (2018) is not in the reference list.

A general thought about trace elements in shell carbonate that was not discussed: Bender and Morse (1990) consider distribution coefficients to be phenomenological by nature; they depend a lot on the aqueous chemistry of the fluids in which carbonate grows and the nature of the solid phase. They use observations like those reported in Mucci and Morse (1983) who show that more Sr can substitute into calcite when more Mg resides in the crystal lattice as evidence of this. Could the possibility exist that biogenic shell aragonite shares a similar fate?

Best of luck, Chris Romanek

---

## Author Comment (AC1) · 4 Dec 2019

Anonymous Referee #1 **Author refers to line numbers in the Track Changes document.** This was a helpful review. Thank you so much for your time and insight. Your review helped make this a better paper. We feel it is a big step closer to being publication-worthy. Review of the manuscript "Trace elements in mussel shells from the Brazor River, Texas..." The paper addresses environmentally important question potentially suitable for Bio- geosciences. The title is not correct: this work is about Mn essentially, rather than trace elements Many references are incomplete Good point. The title is now revised to be more specific to the novel finding of the shell Mn/Ca-discharge relationship. L10: does it simply mean that Sr correlated with Mn in shells? Yes Introduction: The novelty of this study and motivation behind this work are unclear. Why river Mn flux is important at all? Indirect assessment of this flux via shells is not the easiest way... Good point. Now the novelty is stated on lines 15, 64, 260-265. Methods: How the samples were processes after dissolution; were they filtered? Was the dissolution complete? Now line 101 makes clear that shell sample dissolution was complete in the 2% nitric acid ICP-MS solutions. Eqn 1-3: Unclear why this information is needed The equations have been removed. This frees up space. Sampling (L149-150): The water samples were not filtered and acidified. As such, metal concentration (except probably Ca and Sr) in river water could not be measured and distribution coefficients do not make sense. Moreover, the whole main motivation of this study - reconstruction of Mn flux in the river - becomes compromised. As such, the distribution coefficients oven in Table 3 may not be usable. Now lines 110-112 clarify that the samples were not acidified immediately but were brought to 2% nitric for ICP-MS analysis. They were never filtered. While the VanPlantinga et al. (2017) paper discusses broad relationships between major dissolved ions that are consistent with our data, there are indeed risks of preservation problems in this study for the manganese concentration measurements. Therefore the presentation of dissolved manganese concentrations has been removed. The partition coefficient estimates for Mn/Ca in this study are given an asterisk in Table 3 where the water Mn concentration is assumed to be the median value for Brazos River water from a reliable study, Keeney-Kennicutt and Presley (1986).

Keeney-Kennicutt, W.L. and Presley, B.J.: The geochemistry of trace metals in the Brazos River estuary. Estuarine, Coastal and Shelf Science, 22(4), pp.459–477, 1986.

VanPlantinga, A.A., Grossman, E.L. and Roark, E.B.: Chemical and isotopic tracer evaluation of water mixing and evaporation in a dammed Texas river during drought. River Research and Applications, 33(3), pp.450–460, 2017.

In Fig 3C, use log scale for discharge. How good is this correlation? What about correlation with temperature? Done L254-259: Another issue is what is Mn concentration in the lake hypolimnion? If the lake is seasonally stratified, then, during the overturn, the bottom Mn-rich waters can feed the river thus dramatically increasing the Mn concentration in the river water. Lines 287-291 now discuss this issue in the "point source theory" part of the discussion. L257-258: The argument is unclear. In Fig. 3E, Mn/Ca is not inversely related to Sr/Ca. Please show the relationships. This section was weak and needed to be revised. Thank you for directing my attention here. L270-271: The low flow may provide enhanced Mn2+ input from the riparian and hy- porheic zone We now include a reference to the relevant surface-groundwater interaction (Rhodes et al., 2017), which builds on concepts in VanPlantinga et al. (2017). The tracers for bank storage (high [Mg2+ + Ca2+ + HCO3-] relative to [Na+ + Cl-]) are prominent during the 1-2 day period following a rain storm, when bank storage quickly discharges. The idea that the Brazos River is a gaining stream in times of low flow, thus possibly changing the redox related dissolved ion chemistry such as Mn/Ca, is not supported by these studies. Rhodes, K.A., Proffitt, T., Rowley, T., Knappett, P.S., Montiel, D., Dimova, N., Tebo, D. and Miller, G.R.: The importance of bank storage in supplying baseflow to rivers flowing through compartmentalized, alluvial aquifers. Water Resources Research, 53(12), pp.10539-10557, 2017.

L277: notice here that the maximal suspended load is usually observed at high discharge Yes, you read that correctly. However, there is a big emphasis on inorganic particles, being at higher concentrations in more turbid conditions which the mussels are less likely to feed on because this is a strong environmental stressor for them. As our study finds, the shell Mn/Ca is inversely related to log of discharge, so it cannot be the inorganic Mn-bearing particles that the mussels are eating. L288: Chl a of mg/L concentration is really high. May be a misprint here and the concentrations are in $\mu$g/L? This line now specifies that it is water column chlorophyl in mg x L-3 and benthic chlorophyll in mg x L-2 L293-295: As a conclusion to section 3.5, this is extremely discouraging. It looks like one cannot yet discuss the sources of Mn for shells, so this section is useless... We feel that calling for future research into spatial and temporal scales of manganese flux in rivers such as the Brazos River is appropriate. The research from Roach (2014) mentioned in this section gives us a basis to argue our hypothesis that the environmental Mn is ingested by mussels as organic particle bound Mn.

L312 Owen1996 is not in the ref list This has been fixed. Thank you. L325-329: May be place this information in the Introduction. Again, this sentence is very discouraging: how is it related to particulate case analyzed in this work? What is more important, according to authors, in Brazes River: physiological mechanisms or environmental factors? The relative contributions to shell trace element chemistry from physiological factors such as metabolism, ion channels, reproduction vs. environmental factors such as river chemistry is the backdrop of this research. The first paragraph of the introduction covers this issue. I added a "does not necessarily…" here. Rewrite L 347-348 This sentence in the conclusions has now been revised. L349-350: This is not sufficiently discussed and the whole story of DMn can be com- promised by inadequate sampling The partition coefficient estimates for Mn/Ca in this study are given an asterisk in Table 3 where the water Mn concentration is assumed to be the median value for Brazos River water from a reliable study, Keeney-Kennicutt and Presley (1986).

Anonymous Referee #1

**\*\*Author refers to line numbers in the Track Changes document.\*\* This was a helpful review.
Thank you so much for your time and insight. Your review helped make this a better paper. We feel
it is a big step closer to being publication-worthy.**

Review of the manuscript "Trace elements in mussel shells from the Brazor River, Texas..."

The paper addresses environmentally important question potentially suitable for Bio-
geosciences.

The title is not correct: this work is about Mn essentially, rather than trace elements Many
references are incomplete

**Good point. The title is now revised to be more specific to the novel finding of the shell
Mn/Ca-discharge relationship.**

L10: does it simply mean that Sr correlated with Mn in shells?

**Yes**

Introduction: The novelty of this study and motivation behind this work are unclear.
Why river Mn flux is important at all? Indirect assessment of this flux via shells is not the easiest
way...
**Good point. Now the novelty is stated on lines 15, 64, 260-265.**

Methods: How the samples were processes after dissolution; were they filtered? Was the
dissolution complete?

**Now line 101 makes clear that shell sample dissolution was complete in the 2% nitric acid
ICP-MS solutions.**

Eqn 1-3: Unclear why this information is needed

**The equations have been removed. This frees up space.**

Sampling (L149-150): The water samples were not filtered and acidified. As such, metal
concentration (except probably Ca and Sr) in river water could not be measured and distribution
coefficients do not make sense. Moreover, the whole main motivation of this study -
reconstruction of Mn flux in the river - becomes compromised. As such, the distribution
coefficients oven in Table 3 may not be usable.

**Now lines 110-112 clarify that the samples were not acidified immediately but were
brought to 2% nitric for ICP-MS analysis. They were never filtered. While the
VanPlantinga et al. (2017) paper discusses broad relationships between major dissolved**

**Fig. 1.** responses in bold

**Mussel shell Mn/Ca as a novel proxy for discharge in the Brazos**
**River, Texas**

**Authors:** Alexander A. VanPlantinga[1] and Ethan L. Grossman[1]
[1]Department of Geology and Geophysics, Texas A&M University, College Station, Texas, USA 77843-3115

**Abstract.** In sclerochronology, understanding the drivers of shell chemistry is necessary in order to use shells to
reconstruct environmental conditions. We measured the Mg, Ca, Sr, Ba, and Mn contents in water samples and in
the shells of two freshwater mussels (*Amblema plicata* and *Cyrtonaias tampicoensis*) from the Brazos River, Texas
to test their reliability as environmental archives. Shells were analyzed along growth increments using age models
established with stable and clumped isotopes. Shells were also examined with cathodoluminescence (CL)
microscopy to map Mn/Ca distribution patterns. In the shells, Sr/Ca correlated with Mn/Ca, while Mg/Ca and Ba/Ca
showed no clear trends. Mn/Ca correlated inversely with the log of river discharge. Because suspended chlorophyll
concentration is high  during low flow and
suspended inorganic particles (turbidity) is high during high flow, peak Mn/Ca values may come from elevated
feeding or metabolic rates related to the abundance or suspended particulate organic matter. For the first time, shell
Mn/Ca values were used to reconstruct river discharge patterns, which, to our knowledge, has previously only been
performed with shell chemistry using oxygen isotopes.

**1 Introduction**

Sclerochronology is the study of the physical and chemical properties of invertebrate hard parts and the
temporal context in which they grew. It is useful in marine paleoclimatology, but can also be applied to freshwater
ecosystems. There is great potential for using mollusks to reconstruct environmental conditions in the present and in
the geologic past, but problems remain in understanding the relationship between mollusk shell chemistry and the
ambient environment (Immenhauser et al., 2016). For example, shell Sr/Ca can record temperature as a reflection of
mollusk metabolic response to seasonal temperature variation opposite what is thermodynamically predicted for
aragonite (Wheeler, 1992; Gillikin et al., 2005; Carré et al., 2006; Sosdian et al., 2006; Gentry et al., 2008). Shell
Mg/Ca can record temperature (Freitas et al., 2006), and shell Ba/Ca sometimes correlates with diatom primary
productivity (Vander Putten et al., 2000; Lazareth et al., 2003), but it can also be controlled by growth rate (Izumida

**Fig. 2.** track changes

**Mussel shell Mn/Ca as a novel proxy for discharge in the Brazos**
**River, Texas**

**Authors:** Alexander A. VanPlantinga[1] and Ethan L. Grossman[1]
[1]Department of Geology and Geophysics, Texas A&M University, College Station, Texas, USA 77843-3115

**Abstract.** In sclerochronology, understanding the drivers of shell chemistry is necessary in order to use shells to reconstruct environmental conditions. We measured the Mg, Ca, Sr, Ba, and Mn contents in water samples and in the shells of two freshwater mussels (*Amblema plicata* and *Cyrtonaias tampicoensis*) from the Brazos River, Texas to test their reliability as environmental archives. Shells were analyzed along growth increments using age models established with stable and clumped isotopes. Shells were also examined with cathodoluminescence (CL)

microscopy to map Mn/Ca distribution patterns. In the shells, Sr/Ca correlated with Mn/Ca, while Mg/Ca and Ba/Ca showed no clear trends. Mn/Ca correlated inversely with the log of river discharge. Because suspended chlorophyll concentration is high during low flow and suspended inorganic particles (turbidity) is high during high flow, peak

Mn/Ca values may come from elevated feeding or metabolic rates related to the abundance or suspended particulate organic matter. For the first time, shell Mn/Ca values were used to reconstruct river discharge patterns, which, to our knowledge, has previously only been performed with shell chemistry using oxygen isotopes.

**1 Introduction**

Sclerochronology is the study of the physical and chemical properties of invertebrate hard parts and the temporal context in which they grew. It is useful in marine paleoclimatology, but can also be applied to freshwater ecosystems. There is great potential for using mollusks to reconstruct environmental conditions in the present and in the geologic past, but problems remain in understanding the relationship between mollusk shell chemistry and the ambient environment (Immenhauser et al., 2016). For example, shell Sr/Ca can record temperature as a reflection of mollusk metabolic response to seasonal temperature variation opposite what is thermodynamically predicted for aragonite (Wheeler, 1992; Gillikin et al., 2005; Carré et al., 2006; Sosdian et al., 2006; Gentry et al., 2008). Shell

Mg/Ca can record temperature (Freitas et al., 2006), and shell Ba/Ca sometimes correlates with diatom primary productivity (Vander Putten et al., 2000; Lazareth et al., 2003), but it can also be controlled by growth rate (Izumida et al., 2011). Mollusk soft tissue reflects variations in metal bioaccumulation by organ and by element (Arifin and

Bendell-Young, 2000; Chale, 2002; Ravera et al., 2003; Silva et al., 2006; Bellotto and Mieckeley, 2007). Soft tissue

**Fig. 3.** draft without track changes

---

## Author Comment (AC2) · 4 Dec 2019

**\*\*Author refers to line numbers in the Track Changes document.\*\* This was a helpful review. Thank you so much for your time and insight. Your review helped make this a better paper. We feel it is a big step closer to being publication-worthy.**

Review of the manuscript "Trace elements in mussel shells from the Brazor River, Texas..."

The paper addresses environmentally important question potentially suitable for Bio-geosciences.

The title is not correct: this work is about Mn essentially, rather than trace elements Many references are incomplete

**Good point. The title is now revised to be more specific to the novel finding of the shell Mn/Ca-discharge relationship.**

L10: does it simply mean that Sr correlated with Mn in shells?

**Yes**

Introduction: The novelty of this study and motivation behind this work are unclear.
Why river Mn flux is important at all? Indirect assessment of this flux via shells is not the easiest way...
**Good point. Now the novelty is stated on lines 15, 64, 260-265.**

Methods: How the samples were processes after dissolution; were they filtered? Was the dissolution complete?

**Now line 101 makes clear that shell sample dissolution was complete in the 2% nitric acid ICP-MS solutions.**

Eqn 1-3: Unclear why this information is needed

**The equations have been removed. This frees up space.**

Sampling (L149-150): The water samples were not filtered and acidified. As such, metal concentration (except probably Ca and Sr) in river water could not be measured and distribution coefficients do not make sense. Moreover, the whole main motivation of this study - reconstruction of Mn flux in the river - becomes compromised. As such, the distribution coefficients oven in Table 3 may not be usable.

**Now lines 110-112 clarify that the samples were not acidified immediately but were brought to 2% nitric for ICP-MS analysis. They were never filtered. While the VanPlantinga et al. (2017) paper discusses broad relationships between major dissolved**

ions that are consistent with our data, there are indeed risks of preservation problems in this study for the manganese concentration measurements. Therefore the presentation of dissolved manganese concentrations has been removed. The partition coefficient estimates for Mn/Ca in this study are given an asterisk in Table 3 where the water Mn concentration is assumed to be the median value for Brazos River water from a reliable study, Keeney-Kennicutt and Presley (1986).

Keeney-Kennicutt, W.L. and Presley, B.J.: The geochemistry of trace metals in the Brazos River estuary. *Estuarine, Coastal and Shelf Science*, *22*(4), pp.459–477, 1986.

VanPlantinga, A.A., Grossman, E.L. and Roark, E.B.: Chemical and isotopic tracer evaluation of water mixing and evaporation in a dammed Texas river during drought. *River Research and Applications*, *33*(3), pp.450–460, 2017.

In Fig 3C, use log scale for discharge. How good is this correlation? What about correlation with temperature?

**Done**

L254-259: Another issue is what is Mn concentration in the lake hypolimnion? If the lake is seasonally stratified, then, during the overturn, the bottom Mn-rich waters can feed the river thus dramatically increasing the Mn concentration in the river water.

**Lines 287-291 now discuss this issue in the "point source theory" part of the discussion.**

L257-258: The argument is unclear. In Fig. 3E, Mn/Ca is not inversely related to Sr/Ca. Please show the relationships.

**This section was weak and needed to be revised. Thank you for directing my attention here.**

L270-271: The low flow may provide enhanced Mn2+ input from the riparian and hy- porheic zone

**We now include a reference to the relevant surface-groundwater interaction (Rhodes et al., 2017), which builds on concepts in VanPlantinga et al. (2017). The tracers for bank storage (high [$Mg^{2+}$ + $Ca^{2+}$ + $HCO_3^-$] relative to [$Na^+$ + $Cl^-$]) are prominent during the 1-2 day period following a rain storm, when bank storage quickly discharges. The idea that the Brazos River is a gaining stream in times of low flow, thus possibly changing the redox related dissolved ion chemistry such as Mn/Ca, is not supported by these studies.**

**Rhodes, K.A., Proffitt, T., Rowley, T., Knappett, P.S., Montiel, D., Dimova, N., Tebo, D. and Miller, G.R.: The importance of bank storage in supplying baseflow to rivers flowing through compartmentalized, alluvial aquifers. Water Resources Research,**

**53(12), pp.10539-10557, 2017.**

L277: notice here that the maximal suspended load is usually observed at high dis- charge

**Yes, you read that correctly. However, there is a big emphasis on inorganic particles, being at higher concentrations in more turbid conditions which the mussels are less likely to feed on because this is a strong environmental stressor for them. As our study finds, the shell Mn/Ca is inversely related to log of discharge, so it cannot be the inorganic Mn-bearing particles that the mussels are eating.**

L288: Chl a of mg/L concentration is really high. May be a misprint here and the concentrations are in μg/L?

**This line now specifies that it is water column chlorophyl in mg x L$^{-3}$ and benthic chlorophyll in mg x L$^{-2}$**

L293-295: As a conclusion to section 3.5, this is extremely discouraging. It looks like one cannot yet discuss the sources of Mn for shells, so this section is useless...

**We feel that calling for future research into spatial and temporal scales of manganese flux in rivers such as the Brazos River is appropriate. The research from Roach (2014) mentioned in this section gives us a basis to argue our hypothesis that the environmental Mn is ingested by mussels as organic particle bound Mn.**

L312 Owen1996 is not in the ref list
**This has been fixed. Thank you.**

L325-329: May be place this information in the Introduction. Again, this sentence is very discouraging: how is it related to particulate case analyzed in this work? What is more important, according to authors, in Brazes River: physiological mechanisms or environmental factors?

**The relative contributions to shell trace element chemistry from physiological factors such as metabolism, ion channels, reproduction vs. environmental factors such as river chemistry is the backdrop of this research. The first paragraph of the introduction covers this issue. I added a "does not necessarily…" here.**

Rewrite L 347-348

**This sentence in the conclusions has now been revised.**

L349-350: This is not sufficiently discussed and the whole story of DMn can be com- promised by inadequate sampling

**The partition coefficient estimates for Mn/Ca in this study are given an asterisk in Table 3 where the water Mn concentration is assumed to be the median value for Brazos River water from a reliable study, Keeney-Kennicutt and Presley (1986).**

**Anonymous Referee #2**

**\*\*Author refers to line numbers in the Track Changes document.\*\* This was a helpful review. Thank you so much for your time and insight. Your review helped make this a better paper. We feel it is a big step closer to being publication-worthy.**

Alexander A. VanPlantinga & Ethan L. Grossman prepared an interesting work which aimed to provide a better understanding of how environmental and biological factors affect the trace elemental composition in freshwater mussel shells. Yet, my feeling is that the authors may rephrase the title as proposed before, for example, focusing on the application of freshwater mussel shell trace elements to retrospective monitoring of riverine discharge. If the structure of the work went this way, the authors may likely shy away of one of the major flaws of this work – water samples were not filtered and acidified thereby hindering the reliable calculation of distribution coefficient. Actually, I think this parameter is not closely related to the story, as there are already several studies which reported such data even in freshwater bivalve species (Mg, Sr, Mn as well as Ba). As such, I would urge the authors to consider these suggestions, especially given that these results are very interesting.

**The title has been revised to reflect the novelty and focus of the paper. Now the novelty is stated on lines 15, 64, and 264.**

**Now lines 109-112 clarify that the samples were not acidified immediately but were brought to 2% nitric for ICP-MS analysis. They were never filtered. While the VanPlantinga et al. (2017) paper discusses broad relationships between major dissolved ions that are consistent with our data, there are indeed risks of preservation problems in this study for the manganese concentration measurements. Therefore the presentation of dissolved manganese concentrations has been removed. The partition coefficient estimates for Mn/Ca in this study are given an asterisk in Table 3 where the water Mn concentration is assumed to be the median value for Brazos River water from a reliable study, Keeney-Kennicutt and Presley (1986).**

**Keeney-Kennicutt, W.L. and Presley, B.J.: The geochemistry of trace metals in the Brazos River estuary. *Estuarine, Coastal and Shelf Science*, 22(4), pp.459–477, 1986.**

**VanPlantinga, A.A., Grossman, E.L. and Roark, E.B.: Chemical and isotopic tracer evaluation of water mixing and evaporation in a dammed Texas river during drought. *River Research and Applications*, 33(3), pp.450–460, 2017.**

Several minor comments listed below: Introduction – should be rephrased to give a clear clue of the work. Line 17, The definition of sclerochronology is not complete, simply referring to Wikipedia and references therein. Line 56, why did the authors expect a relationship between Sr/Ca and temperature?

**The definition of sclerochronology has been revised. The first paragraph of the introduction brings up the shell Sr/Ca - temperature relationship observed in many studies.**

Method Lines 99-101, I appreciate that authors are not trying to hide this flaw in the manuscript but this indeed strongly limits the strength of conclusions.

**The water chemistry data has been simplified in Figure 4A and 4B. The corresponding discussion has been simplified in lines 155-170.**

**The partition coefficient estimates for Mn/Ca in this study are given an asterisk in Table 3 where the water Mn concentration is assumed to be the median value for Brazos River water from a reliable study, Keeney-Kennicutt and Presley (1986).**

Results & Discussion Lines 142-143, if the authors reported and discussed trace el- emental concentrations in shells in the form of element-to-calcium ratio (Me/Ca), so please stay consistent throughout the manuscript. In fact, calcium concentration in bi- valve shells can most likely be consistent over time. Line 173, in the leading sentence, if you are talking about "significantly", "p" value should always be given. Section 3.6, in my opinion, the authors should devote efforts to discussing the potential of Mn/Ca as an indicator of riverine discharge.

**Noted. It has been revised for consistency. The "p" values have been emphasized throughout, for example lines 200 and 251 and the Figure 4 caption.**

Conclusion The leading two sentences are nicely written to give a clear summary of the work. In line 345, removing Roach et al. (2014).

**It has been removed.**

**Chris Romanek (Referee)**

christopher.romanek@gmail.com

**\*\*Author refers to line numbers in the Track Changes document.\*\* This was a helpful review. Thank you so much for your time and insight. Your review helped make this a better paper. We feel it is a big step closer to being publication-worthy.**

Review of bg-2019-323: Trace elements in mussel shells from the Brazos River, Texas: environmental and biological control by VanPlantinga and Grossman

This manuscript describes an investigation to better understand the trace element geo- chemistry of two freshwater mussel shells collected from the Brazos River TX in 2013. The investigators serially sampled the inner nacreous layer (INL) and outer ventral mar- gin (VM) of a shell of specimen of Amblema plicata and Cyrtonaias tampicoensis and they analyzed the carbonate for Mg, Ca, Sr, Ba and Mn concentration. The data were placed within a temporal framework using previous oxygen isotope data (VanPlantinga and Grossman, 2018) and compared to other environmental data (e.g., temperature, water chemistry, water discharge rate) to better understand the origin(s) of the geo- chemical signatures locked in the shell. The principal outcome of the study focused on shell Mn/Ca records which correlated inversely with river discharge allowing for a reconstruction of river discharge patterns. The investigators conclude that Mn was attained primarily through the ingestion of Mn-bearing particulate organic matter.

This manuscript addresses subject matter that is of general interest to the Biogeo- science community and it identifies a novel proxy of river discharge in the geochemical record of the shell. The investigators thoughtfully consider and evaluate the relevant peer reviewed literature within the context of their study.

The manuscript is relatively well written, although I had some trouble with the PDF because the figures contained a lot of information that could not be seen easily on a paper copy.

The amount of information provided on shell geochemistry and the various correlations and comparisons made me feel overwhelmed at times. The Mg, Sr, and Ba records (as well as ï ˛Ad'13Cand CL) appear ancillary at times and it is unclear if they provide meaningful supporting evidence for the major conclusions of the manuscript.

**We agree. The Mg, Sr, and Ba content has been condensed to make room for the focus of the study.**

I recommend that the investigators reconsider what is really needed to support the main conclusion(s) of their study. This should help to sharpen the take home message of the study. A more highly focused revision that centers on the Mn records would be valuable to the scientific community and make a meaningful contribution to the pub- lished literature. Data that may be

excluded from a revision could likely form the basis for separate manuscript(s).

Other comments, identified by line number (L), are provided below: General comment: Each time a geochemical profile is presented or discussed the investigators should identify whether it is from the inner nacreous layer or the ventral margin. Sometimes this distinction is clearly made while at other times it appears ambiguous.

**I went back and added some clarification on lines 200, 227, and 251. The real focus is on the ventral margin, but some comparison between ONL and VM/INL values is offered.**

L17: The first sentence that defines the word "sclerochronology" is inaccurate. Scle-rochronology is more specific than "... the study of the physical and chemical prop- erties of invertebrate hard parts...", it involves the temporal context in which these properties are considered.
**Good catch. Line 21, in the introduction, now has the full definition.**

L59: Methods: The physicochemical water sampling procedures need to be explained in greater detail, or cited properly. This extends to L99-101 where the statement is made that "... water samples were not filtered and acidified for analysis after months in storage." The implications of this unusual sampling strategy should be explained in greater detail. How severely is the Mn data compromised? How are other elements affected in addition to Mn?

**Lines 111, 161, 306-308 now mention the likely fate of sampled dissolved Mn, adsorption to inorganic particles.**

**Now lines 110-112 clarify that the samples were not acidified immediately but were brought to 2% nitric for ICP-MS analysis. They were never filtered. While the VanPlantinga et al. (2017) paper discusses broad relationships between major dissolved ions that are consistent with our data, there are indeed risks of preservation problems in this study for the manganese concentration measurements. Therefore the presentation of dissolved manganese concentrations has been removed. The partition coefficient estimates for Mn/Ca in this study are given an asterisk in Table 3 where the water Mn concentration is assumed to be the median value for Brazos River water from a reliable study, Keeney-Kennicutt and Presley (1986).**

**Keeney-Kennicutt, W.L. and Presley, B.J.: The geochemistry of trace metals in the Brazos River estuary. *Estuarine, Coastal and Shelf Science*, 22(4), pp.459–477, 1986.**

**VanPlantinga, A.A., Grossman, E.L. and Roark, E.B.: Chemical and isotopic tracer evaluation of water mixing and evaporation in a dammed Texas river during drought. *River Research and Applications*, 33(3), pp.450–460, 2017.**

L74: Hydrogen isotope compositions are not discussed in the text so a description of sample/analysis procedures in not warranted. Also, anytime delta notation is used, e.g., ï

˛Ad'18O, the word "value" should follow it.

**The hydrogen isotopes mention has been removed.**

L90 and L94: It appears the sampling resolution for stable isotopes (60 ˛ ˛A g) and ICP-MS analysis (20-160 ˛ ˛A g) differed. The implications of this should be addressed somewhere in the text.

**The intervals are the same. The stable isotopes and trace element data come from paired sampling for ICP-MS and IRMS analyses, so in lines 100-102, I clarify this.**

L102: Were the CL images taken before or after the sampling of the shell for isotopic and elemental analysis? The observation of "shadows" should be explained a little better. This could be done in a better description of the CL imaging in general. What is the ultimate purpose of CL imaging? It appears the investigators wished to correlate brightness to measured Mn concentration in the shell. Was the sampling resolution for ICP-MS comparable to the width of bright and dark bands in the CL images?

**The CL images were taken after the micromill samples were taken as now specified in line 113. This is now clarified on lines. CL can theoretically be done independently of the shell ICP/IRMS analyses by using multiple adjacent thin isomet slices of the shell.**

**I honestly do not know where the shadows come from. They were a nuisance.**

**Lines 114-115 now clarify the motive of using CL.**

**Lines 268-269 now specify the scale as being at a resolution finer than 1mm for the CL images, comparable with micromilling.**

L121: Oxygen isotopes. The first paragraph can be reduced if the manuscript needs to be shortened.

**The equations have been removed.**

L146-148: Variance in a data set is independent of the scale over which the data are considered, e.g., the Mg data set (12 ppb – 20 ppm) is more variable than the Ca dataset (19-83 ppm) regardless of whether the data are considered on a linear or log scale. Also, reference to Fig. 4 (L148) is for Me/Ca ratios and not absolute concentrations.

**The water chemistry plots (now Figures 4A and 4B) now show meaningful trends attributable to well characterized processes from VanPlantinga et al. (2017) and Chowdhury et al. (2010). Hopefully this makes the water chemistry discussion easier for the reader.**

**Chowdhury A, Osting T, Furnans J, Mathews R.: Groundwater–surface water interaction in the Brazos River Basin: evidence from lake connection history and chemical and isotopic compositions: Texas Water Development Board Report, 375 (August):1–61, 2010.**

**VanPlantinga, A.A., Grossman, E.L. and Roark, E.B.: Chemical and isotopic tracer evaluation of water mixing and evaporation in a dammed Texas river during drought. *River Research and Applications*, *33*(3), pp.450–460, 2017.**

L155: The word "shell" should proceed the word "growth".

**Done**

L162: DMg values reported in Table 3 range from 1-138 (10-3) or 0.001-0.0138, not "0.001-0.138" as stated in the text.

**The orders of magnitude are correct actually. The revised D values for Mg changed slightly in the new draft because the water chemistry was not being taken from the same consistent dates for all trace elements. I fixed that. It has no impact on the findings.**

L163: "Mg/Ca does not show any [temporal] systematic trend in our water data . . ."

**The word "temporal" has been added.**

L190: Sentence starting with "Ba/Ca values . . ." is redundant given L186 . L211: Vari- ability in shell Mn/Ca in not reported in Table 1.

**Okay. That has been fixed in the process of condensing down the Ba, Sr, and Mg discussion.**

L212: I think the reference should be to "Figure 3B" here and not "Figure 4B".

**Yes, we mixed up Figures 3 and 4 and reversed them again in the forthcoming revisions.**

L237: The final statement in the paragraph is introductory and should come earlier in the manuscript.

**The novel inverse Mn/Ca-logQ relationship is highlighted in the revised title and the introduction in line 64.**

L238: The section entitled Cathodoluminescence is too brief. It should be expanded and integrated better in the text.

**The main purpose of the CL discussion paragraph is to point out that the Mn/Ca we measured represented lattice-bound Mn/Ca rates. It seemed like stating that more than once would be repetitive.**

L260: Is this the only place where the carbon isotope data are discussed except for L184-185? The carbon isotope data should be integrated more fully in the text, e.g., why are there correlations between shell growth and Sr/Ca vs ï¸Ad'13C values?

**The vague relationships between shell Mn/Ca, Sr/Ca, and carbon isotopes seen in one**

**shell or the other suggests a metabolic link, but the fact that it is d13C-Mn/Ca in one shell and d13C-Sr/Ca in another shell makes it too vague. All I can do is mention the Lake Whitney - shell d13C relationship from VanPlantinga and Grossman (2018).**

L283: Goodwin et al. (2018) is not in the reference list.

**Added it.**

A general thought about trace elements in shell carbonate that was not discussed: Bender and Morse (1990) consider distribution coefficients to be phenomenological by nature; they depend a lot on the aqueous chemistry of the fluids in which carbonate grows and the nature of the solid phase. They use observations like those reported in Mucci and Morse (1983) who show that more Sr can substitute into calcite when more Mg resides in the crystal lattice as evidence of this. Could the possibility exist that biogenic shell aragonite shares a similar fate?
Best of luck, Chris Romanek

**Interesting point. I do not really have a great answer. My response would be informed by Soldati et al. (2016).**

> **Soldati, A.L., Jacob, D.E., Glatzel, P., Swarbrick, J.C. and Geck, J.: Element substitution by living organisms: the case of manganese in mollusc shell aragonite. *Scientific reports*, 6, p.22514, 2016.**

---

## Author Comment (AC3) · 4 Dec 2019

**Mussel shell Mn/Ca as a novel proxy for discharge in the Brazos River, Texas**

**Authors:** Alexander A. VanPlantinga[1] and Ethan L. Grossman[1]

[revised manuscript text omitted]

and δ¹⁸O$_{water}$ to predict shell δ¹⁸O according to equations 1, 2, and 3 (Dettman et al.,1999, based on Grossman and

Ku, 1986).

[Figure]

$$\ln (\alpha \; {}^{aragonite}_{water}) = 2.559 \; x \; (10^6 \; x \; T^{-2}) + 0.715 \qquad (1)$$

$$\alpha \; {}^{aragonite}_{water} = \frac{(1000 + \delta^{18}O_{aragonite} VPDB)}{(1000 + \delta^{18}O_{water} VSMOW)} \qquad (2)$$

$$\alpha \; {}^{VSMOW}_{VPDB} = 1.0309 \text{ (Gonfiantini et al., 1995).} \qquad (3)$$

B Because winter hiatuses and erratic summer growth patterns result in chaotic shell δ¹⁸O patterns that complicate

δ¹⁸O sclerochronology, we used clumped isotope thermometry to supplement δ¹⁸O data (VanPlantinga and

Grossman, 2018).

Based on our shell chronology, the time interval represented by the trace element analyses is April to

August 2013. During this interval water temperature and δ¹⁸O$_{water}$ values ranged from 13 to 34°C and -2.7 to 1.3‰, respectively.  Daily averaged river discharge at the study site was 173-2230 cfs (cubic feet per second; USGS gage

08108700; https://waterdata.usgs.gov). The higher δ¹⁸O$_{water}$ values reflect increased summer evaporation combined with increased proportion of flow from evaporated ¹⁸O-enriched Lake Whitney water, whereas lower values (as low as -8‰) are the result of ¹⁸O-depleted precipitation and runoff (Chowdhury et al., 2010; VanPlantinga et al., 2017).

**3.2 Water chemistry**

Mean water Me/Ca values are presented in Table 1. Water dissolved ion concentration and electrical conductivity results are shown in Figures 4A and 4B. The Sr, Ca, and Ba results correlate track with the electrical conductivity (p < 0.05) because Brazos River salinity is strongly controlled by the proportion of river flow discharged from Lake Whitney (Chowdhury et al., 2010; VanPlantinga et al., 2017). The correlation between Mg and Ca (Figure 4B) is due to the dominance of Mg2+ and Ca2+ over Na+, Sr2+, and Ba2+, and Cl- in the runoff and bank storage water source endmember (VanPlantinga et al., 2017; Rhodes et al., 2017). Brazos River Alluvium

Aquifer (BRAA) influence is strongest in the hours after strong rain (Rhodes et al., 2017), and so manganese- scavenging particles in the high-Ca2+-Mg2+-HCO3- samples probably explains the inverse Ca-Mn correlation in our unfiltered water samples, where manganese carbonates are favored. Mg, Sr, and Ba correlated positively with Ca concentrations and Mn correlated negatively with Ca (Rsq > 0.55, p < 0.0007). Water Mn/Ca, Ba/Ca, and Sr/Ca values (mmol/mol) significantly correlate with each other (p < 0.00011), and further, Mg/Ca weakly correlates with

**Commented [1]:** needs clarification, relationships are confusing

**Commented [2]:** Electrical conductivity, as a proxy for LW influence, does not show a relationship with water Mn(ppb), the p value is 0.57, rsq = -0.04.

**Commented [3]:** _Marked as resolved_

**Commented [4]:** _Re-opened_

USGS data for the Brazos River gage at Bryan, Texas (08108700) generally display an inverse relationship between dissolved oxygen and discharge.

While he low measured water manganese concentrations (0.1-0.6 ppb) are consistent with Keeney-Kennicutt and Presley's (1986) measurements of Brazos River water (0.1-

2.3 ppb),  our water samples were not filtered after  acidified immediately  collection, so true dissolved Mn2+ from the time of sampling cannot be discussed. Turekian and Scott (1967) attribute the suspended particulate manganese concentration in the Brazos

River (690 ppm) to soil erosion, as found in other river Mn studies (e.g., Shiller, 2002; Risk et al., 2010).

**3.3 Shell chemistry**

Table 2 explores relationships between environment, shell growth, and shell chemistry using Pearson's *r*

values. Me/Ca values and distribution coefficients ($D_{Me}$) can differ between specimens 3R5 and TP3, and between the ventral margin (VM) and inner nacreous layer (INL) of the same shell, especially with regard to Mg/Ca and

Mn/Ca (Table 1). Nevertheless, taken as a whole, the ranges in values are generally similar to those recorded in previous studies of freshwater mussels (Carroll and Romanek, 2008; Geeza et al., 2018 and references cited therein), except for Mg/Ca (Table 3). In addition, log of shell $D_{metal}$ values overlap with the results in Bolotov et al. (2015) for metal/calcium partitioning in *Margaritifera*, except that their Mg/Ca values are 1-4 orders of magnitude lower than ours (0.001-0.115).

here are no systematic temporal variations in the Mg/Ca or Ba/Ca values of the shells (Figures 3B and 3D). In terms of  taxonomic differences , Mg/Ca values of 3R5 are about three times greater than those of TP3.

Brazos River water Mg/Ca is about half that in the Scioto River in Ohio (Geeza et al., 2018), but our average shell Mg/Ca values are nearly an order of magnitude higher, resulting in

**Commented [5]:** can we still give a rough D(Mn/Ca) estimate combining our shell data with the median of their published data (0.1-2.3ppb)???

**Commented [6]:** _Marked as resolved_

**Commented [7]:** _Re-opened_

**Commented [8]:** confusing. Contradicts Figure 3B

[revised manuscript text omitted]
 and experiences a fall overturn where manganese and iron-rich oxygen depleted water mixes with the overlying water column (Strause and Andrews, 1984), but the temperature and water chemistry changes in the spring and summer are typically more gradual. Hydroelectric releases from the Whitney dam flow from 7 meters above lake bottom. The hypothesis that seasonal stratification patterns in Lake Whitney,

240 km away, drive downstream mussel shell Mn/Ca variation in the spring is not supported. Elevated $\delta^{13}C$ in the shells during the summer of 2013 was interpreted as an indication of heightened Lake Whitney influence on river flow and chemistry during drought conditions (VanPlantinga and Grossman, 2018; VanPlantinga et al., 2017). There is a correlation between $\delta^{13}C$ and Mn/Ca in 3R5 but not in TP3. There is not yet sufficient evidence to indicate that

Lake Whitney or the Little River are point sources of shell manganese, nor to explain the striking inverse shell

Mn/Ca - river discharge relationship, but this does not diminish

**Commented [10]:** water Mn/Ca positively correlates with water Sr/Ca and Ba/Ca but not Mg/Ca

**Commented [11]:** there is no relationship between river Ca(ppb) and ECond

**Commented [12]:** If this is the argument for Mn source, then it must be plotted the important role Lake Whitney plays in  major dissolved ion chemistry downstream (VanPlantinga et al., 2017).

Dissolved $Mn^{2+}$ is the most bioavailable form of manganese (Campbell, 1995). Shell Mn/Ca values have been attributed to variations in dissolved $Mn^{2+}$ in the water column (Frietas et al., 2006; Barats et al., 2008) and in the sediment porewater (Zhao et al., 2017a). As mentioned earlier, experimental studies have confirmed that dissolved $Mn^{2+}$ content is recorded in shell Mn/Ca (Jeffree et al., 1995; Hawkes et al., 1996; Markich et al., 2002;

Langlet et al., 2006; Lartaud et al., 2010). However, the low dissolved oxygen conditions in the Brazos River, which should increase the concentration of  dissolved $Mn^{2+}$, occur at times of high flow (USGS 08108700 gage data from waterdata.usgs.gov) when shell Mn/Ca is relatively low. These data related to redox conditions in the water column do not explain the shell Mn/Ca patterns, and we lack the data to evaluate the hypothesis that sediment porewater drives shell Mn/Ca, or if HCO3--rich bank water favors the precipitation of dissolved Mn2+ out of solution in the days following heavy rainfall. These hypotheses should be explored in future studies.

Particulate Mn, bound to organic or inorganic particles, can also be a source of Mn in shells. The inverse relationship between water Ca and Mn concentrations (Figure 4B) indicates that Mn solubility may be related to runoff and rapidly discharged bank storage from local rain storms (Rhodes et al., 2017)

[revised manuscript text omitted]

Figure 4. (A) Lake Whitney tracer parameters for water samples collected from Brazos River (2012-2013) in Bryan-College Station, TX. (B) Tracers that vary systematically in relation to event flow from runoff and bank storage in the same water samples (VanPlantinga et al., 2017). (C) Discharge vs. Mn/Ca. (D) Temperature vs. Sr/Ca. (E) Shell Sr/Ca vs. shell Mn/Ca values. (F) Log10 of river discharge (Q) and reconstructions of log10 (Q) based on the shell Mn/Ca-Q relationship. All R squared (rsq) $p$ values < 0.05.

[Figure]

**Figure 3.** (A) Water chemistry measurements from the Brazos River (2012-2013); empty squares = Mn (ppb), empty circles = Ba (log10 of ppb), empty upright triangles = Ca (log10 of ppb), filled squares = electrical conductivity (log10 of μS), inverted empty triangles = Mg (log10 of ppb), diamonds = Sr (log10 of ppb). (B) Water Mn/Ca (100*mmol/mol) and shell INL Mn/Ca in mmol/mol. (C) Discharge vs. Mn/Ca. (D) Temperature vs. Sr/Ca. (E) Shell Sr/Ca vs. shell Mn/Ca values. (F) Log10 of river discharge (Q) and reconstructions of log10 (Q) based on the shell Mn/Ca-Q relationship.

[Figure]

**Figure 4.** TP3 (black triangles and/or black line), and 3R5 (gray squares and/or gray line) values for shell Mn/Ca and CL (A); shell Mg/Ca (B); shell Sr/Ca (C); shell Ba/Ca (D); water δ[18]O and temperature (E); estimated shell growth rate (F); shell δ[18]O chronologies for TP3, 3R5, and predicted aragonite δ[18]O (G); and shell δ[13]C chronologies (H). The shell isotope chronologies are described in detail in Van Plantinga and Grossman (2018).

**Table 1.** Summary of MACS3 check standard results, error analysis, Brazos River water and shell chemistry results summarized by shell region for trace metal Me/Ca values and calculated partition coefficients D(Me/Ca of shell/water).

| | Mn/Ca | Sr/Ca | Ba/Ca | Mg/Ca |
|---|---|---|---|---|
| MACS3 standard and uncertainty analysis | | | | |
| Mean* | 1.07 | 8.7 | 0.05 | 7.7 |
| Std. dev.* | 0.082 | 0.187 | 0.004 | 0.151 |
| RSD | 7.60% | 2.20% | 7.60% | 2.00% |
| Precision | 2.20% | 0.60% | 2.20% | 0.60% |
| Accuracy | 3.50% | 0.70% | 6.50% | 3.90% |
| Cert. values* | 1.11 | 8.76 | 0.05 | 8.01 |
| Mean water and shell values (mmol/mol Ca) | | | | |
| Water | 0.006 | 5.45 | 0.46 | 292.9 |
| TR5VM | 0.26 | 0.88 | 0.057 | 5.83 |
| TR5INL | 0.83 | 1.13 | 0.086 | 0.80 |
| TP3VM | 0.44 | 0.82 | 0.074 | 1.91 |
| TP3INL | 1.25 | 1.05 | 0.056 | 14.14 |
| Mean distribution coefficients | | | | |
| TR5VM | 13* | 0.14 | 0.10 | 0.01 |
| TR5INL | 42* | 0.18 | 0.16 | 0.002 |
| TP3VM | 22* | 0.13 | 0.13 | 0.005 |
| TP3INL | 63* | 0.16 | 0.10 | 0.04 |

**Table 1.** Summary of MACS3 check standard results and error analysis and Brazos River water and shell results by shell region for trace metal Me/Ca values and calculated partition coefficients D(Me/Ca of shell/water).

| | Mn/Ca | Sr/Ca | Ba/Ca | Mg/Ca |
|---|---|---|---|---|
| **MACS3 check standard and uncertainty analysis** | | | | |
| Mean* | 1.07 | 8.70 | 0.05 | 7.70 |
| Std. dev.* | 0.082 | 0.187 | 0.004 | 0.151 |
| RSD | 7.6% | 2.2% | 7.6% | 2.0% |
| Precision | 2.2% | 0.6% | 2.2% | 0.6% |
| Accuracy | 3.5% | 0.7% | 6.5% | 3.9% |
| Cert. values* | 1.11 | 8.76 | 0.05 | 8.01 |
| *mmol/mol Ca | | | | |
| | | | | |
| **Mean Brazos River and mussel shell values (mmol/mol Ca)** | | | | |
| Water | 0.006 | 5.45 | 0.46 | 292.9 |
| TR5VM | 0.26 | 0.88 | 0.058 | 6.86 |
| TR5INL | 0.83 | 1.13 | 0.085 | 0.79 |
| TP3VM | 0.44 | 0.82 | 0.072 | 2.07 |
| TP3INL | 1.29 | 1.05 | 0.058 | 13.63 |
| | | | | |
| **Mean distribution coefficients** | | | | |
| TR5VM | 27 | 0.14 | 0.11 | 0.02 |
| TR5INL | 89 | 0.18 | 0.16 | 0.002 |
| TP3VM | 47 | 0.13 | 0.14 | 0.006 |
| TP3INL | 135 | 0.16 | 0.11 | 0.04 |

**Table 2.** $r^2$ and $p$ values for relationships between log10 of discharge (log Q), temperature (T), river water $\delta^{18}O_w$, growth rate (G in mm/month), $\delta^{18}O$, $\delta^{13}C$, Mn/Ca, Sr/Ca and CL for specimens TP3 and 3R5. $R^2$ and $p$ values are in **bold** if $p < 0.001$, black if $p < 0.05$, and gray if $p > 0.05$.

| | CL R² | CL p | Mn/Ca R² | Mn/Ca p | Sr/Ca R² | Sr/Ca p | G R² | G p | δ¹⁸O R² | δ¹⁸O p |
|---|---|---|---|---|---|---|---|---|---|---|
| **TP3** | | | | | | | | | | |
| log Q | **0.31** | **3.7E-04** | **0.48** | **1.6E-06** | 0.13 | 2.6E-02 | 0.02 | 4.2E-01 | 0.20 | 6.1E-03 |
| T | 0.00 | 7.4E-01 | 0.18 | 9.8E-03 | 0.24 | 1.2E-03 | 0.06 | 1.4E-01 | 0.04 | 2.1E-01 |
| δ¹⁸Oₓ | 0.13 | 3.0E-02 | 0.07 | 1.0E-01 | 0.18 | 8.4E-03 | | | **0.56** | **1.2E-07** |
| G | 0.26 | 1.4E-03 | **0.27** | **9.5E-04** | 0.24 | 2.3E-03 | | | | |
| δ¹⁸O | | | 0.06 | 1.6E-01 | 0.12 | 3.7E-02 | | | | |
| δ¹³C | 0.14 | 2.5E-02 | 0.09 | 7.4E-02 | 0.20 | 5.9E-03 | 0.10 | 5.6E-02 | | |
| CL | | | **0.43** | **1.2E-05** | **0.34** | **1.7E-04** | | | | |
| Sr/Ca | | | **0.49** | **1.5E-06** | | | | | | |
| **3R5** | | | | | | | | | | |
| log Q | 0.16 | 2.1E-02 | **0.30** | **2.3E-05** | 0.29 | 1.6E-03 | 0.00 | 9.1E-01 | 0.12 | 5.6E-02 |
| T | 0.18 | 1.5E-02 | 0.27 | 2.3E-03 | 0.38 | 1.2E-03 | 0.00 | 7.4E-01 | 0.03 | 3.1E-01 |
| δ¹⁸Oₓ | 0.02 | 4.6E-01 | 0.17 | 2.0E-02 | **0.53** | **2.0E-06** | | | **0.65** | **2.6E-08** |
| G | 0.21 | 8.4E-03 | 0.04 | 2.9E-01 | 0.01 | 6.4E-01 | | | | |
| δ¹⁸O | | | 0.22 | 7.3E-03 | **0.58** | **4.9E-07** | | | | |
| δ¹³C | 0.20 | 1.1E-02 | 0.25 | 3.2E-03 | **0.53** | **2.7E-06** | 0.06 | 1.8E-01 | | |
| CL | | | **0.61** | **1.6E-07** | 0.31 | 1.0E-03 | | | | |
| Sr/Ca | | | **0.55** | **7.6E-07** | | | | | | |

**Table 2.** $r^2$ and $p$ values for relationships between log10 of discharge (log Q), temperature (T), river water $\delta^{18}O_w$, growth rate (G in mm/month), $\delta^{18}O$, $\delta^{13}C$, Mn/Ca, Sr/Ca and CL for specimens TP3 and 3R5. $R^2$ and $p$ values are in **bold** if $p$ is less than the Bonferroni-corrected α value of 0.05 / 52 = 0.001. Gray italicized $p$ values exceed the Bonferroni-corrected α value.

| | CL R² | CL p | Mn R² | Mn p | Sr R² | Sr p | G R² | G p | δ¹⁸O R² | δ¹⁸O p |
|---|---|---|---|---|---|---|---|---|---|---|
| **TP3** | | | | | | | | | | |
| **log Q** | **0.31** | **3.7E-04** | **0.49** | **1.6E-06** | 0.13 | 2.6E-02 | *0.02* | *4.2E-01* | 0.20 | 6.1E-03 |
| **T** | *0.00* | *7.4E-01* | 0.18 | 9.8E-03 | 0.26 | 1.2E-03 | *0.06* | *1.4E-01* | *0.04* | *2.1E-01* |
| **δ¹⁸Oᵥ** | 0.13 | 3.0E-02 | *0.07* | *1.0E-01* | 0.18 | 8.4E-03 | | | **0.56** | **1.2E-07** |
| **G** | 0.26 | 1.4E-03 | **0.27** | **9.5E-04** | 0.24 | 2.3E-03 | | | | |
| **δ¹⁸O** | | | *0.06* | *1.6E-01* | 0.12 | 3.7E-02 | | | | |
| **δ¹³C** | 0.14 | 2.5E-02 | *0.09* | *7.4E-02* | 0.20 | 5.9E-03 | *0.10* | *5.6E-02* | | |
| **CL** | | | **0.43** | **1.2E-05** | **0.34** | **1.7E-04** | | | | |
| **Sr/Ca** | | | **0.49** | **1.5E-06** | | | | | | |
| **3R5** | | | | | | | | | | |
| **log Q** | 0.16 | 2.1E-02 | **0.45** | **2.3E-05** | 0.29 | 1.6E-03 | *0.00* | *9.1E-01* | *0.12* | *5.6E-02* |
| **T** | 0.18 | 1.5E-02 | 0.27 | 2.3E-03 | 0.30 | 1.2E-03 | *0.00* | *7.4E-01* | *0.03* | *3.1E-01* |
| **δ¹⁸Oᵥ** | *0.02* | *4.6E-01* | 0.17 | 2.0E-02 | **0.53** | **2.0E-06** | | | **0.65** | **2.6E-08** |
| **G** | 0.21 | 8.4E-03 | *0.04* | *2.9E-01* | *0.01* | *6.4E-01* | | | | |
| **δ¹⁸O** | | | 0.22 | 7.3E-03 | **0.58** | **4.9E-07** | | | | |
| **δ¹³C** | 0.20 | 1.1E-02 | 0.25 | 3.2E-03 | **0.53** | **2.7E-06** | *0.06* | *1.8E-01* | | |
| **CL** | | | **0.61** | **1.6E-07** | 0.31 | 1.0E-03 | | | | |
| **Sr/Ca** | | | **0.56** | **7.6E-07** | | | | | | |

Table 3. Comparison of shell chemistry and shell/water distribution coefficient results ($D_{Me}$) with past studies (based on Geeza et al., 2017). The manganese partition coefficient ($D_{Mn}$) was calculated assuming median water manganese concentrations (1.2 ppb) from Keeney-Kennicutt and Presley (1986).

| Reference | Sr (mg/kg) | $D_{Sr}$ | Ba (mg/kg) | $D_{Ba}$ | Mg (mg/kg) | $D_{Mg}$ (×10⁻⁴) | Mn (mg/kg) | $D_{Mn}$ | Dissolved Mn |
|---|---|---|---|---|---|---|---|---|---|
| Faure et al. (1967) | | 0.22–0.28 | | | | | | | |
| Nyström et al. (1996) | 300–600 | | | | | | 10–600 | | |
| Mutvei and Westermark (2001) | | | | | | | 400–6000 | | |
| Markich et al. (2002) | | | | | | | 300–1700 | 0.6 | |
| Verdegaal (2002) | 120–220 | | | 0.1 | | | 100–700 | 0.5 | |
| Bailey and Lear (2006) | 700–1000 | 0.28 | | | | | | | |
| Langlet et al. (2007) | | | | | | | 100–1000 | | |
| Ravera et al. (2007) | | | | | | | 200–800 | | |
| Carroll and Romanek (2008) | 120–2000 | 0.17–0.26 | 60–400 | 0.05 | | | 80–1700 | 0.2–0.5 | 36–188 |
| Izumida et al. (2011) | | 0.18–0.22 | | 0.069–0.08 | 150–500 | 0.30–0.42 | | | |
| Bolotov et al. (2015) | 345–595 | 0.15–0.26 | 32–92 | 0.2–0.6 | 23–43 | 0.2–0.4 | 139–469 | 10–300 | |
| Zhao et al. (2017) | 1130–1380 | | | | | | 400–1800 | | 70–1400 |
| Geeza et al. (2017) | 820–3343 | 0.16–0.20 | 15–270 | 0.11–0.14 | 26–1200 | 0.3–0.8 | 120–1250 | 32–42 | 10–60 |
| This study | 430–5279 | 0.08–0.19 | 45–2748 | 0.06–0.46 | 36–89718 | 1–115 | 67–2308 | 6.39* | 0.1–0.6 |

| Water Data | Sr | | Ba | | Mg | | Mn | | |
|---|---|---|---|---|---|---|---|---|---|
| Water Conc. (ppb), this study | 255–852 | | 34–112 | | 12–20837 | | 0.1–0.6 | | |
| Water Me/Ca (mmol/mol) | 3–10 | | 0.2–1.0 | | 0.7–714.9 | | 0.001–0.022 | | |

Table 3. Comparison of shell chemistry and shell/water distribution coefficient results ($D_{Me}$) with past studies (based on Geeza et al., 2017).

| Reference | Sr (mg/kg) | $D_{Sr}$ | Ba (mg/kg) | $D_{Ba}$ | Mg (mg/kg) | $D_{Mg}$ ($\times10^{-3}$) | Mn (mg/kg) | $D_{Mn}$ | Dissolved Mn |
|---|---|---|---|---|---|---|---|---|---|
| Faure et al. (1967) | | 0.22–0.28 | | | | | | | |
| Nyström et al. (1996) | 300–600 | | | | | | 10–600 | | |
| Mutvei and Westermark (2001) | | | | | | | 400–6000 | | |
| Markich et al. (2002) | | | | | | | 300–1700 | 0.6 | |
| Verdegaal (2002) | 120–220 | | 0.1 | | | | 100–700 | 0.5 | |
| Bailey and Lear (2006) | 700–1000 | 0.28 | | | | | | | |
| Langlet et al. (2007) | | | | | | | 100–1000 | | |
| Ravera et al. (2007) | | | | | | | 200–800 | | |
| Carroll and Romanek (2008) | 120–2000 | 0.17–0.26 | 60–400 | 0.05 | | | 80–1700 | 0.2–0.5 | 36–188 |
| Izumida et al. (2011) | | 0.18–0.22 | | 0.069–0.086 | 150–500 | 0.30–0.42 | | | |
| Bolotov et al. (2015) | 345–595 | 0.15–0.26 | 32–92 | 0.2–0.6 | 23–43 | 0.2–0.4 | 139–469 | 10–300 | |
| Zhao et al. (2017) | 1130–1380 | | | | | | 400–1800 | | 70–1400 |
| Geeza et al. (2017) | 820–3343 | 0.16–0.20 | 15–270 | 0.11–0.14 | 26–1200 | 0.3–0.8 | 120–1250 | 32–42 | 10–60 |
| This study | 430–5279 | 0.08–0.19 | 45–2748 | 0.06–0.47 | 36–89718 | 1–138 | 67–2308 | 13–84 | 0.1–0.6 |
| Water Data | Sr | | Ba | | Mg | | Mn | | |
| Water Conc. (ppb), this study | 255–852 | | 34–112 | | 12–20837 | | 0.1–0.6 | | |
| Water Me/Ca (mmol/mol) | 3–10 | | 0.2–1.0 | | 0.7–714.9 | | 0.001–0.022 | | |

---

## Author Comment (AC4) · 4 Dec 2019

The comment was uploaded in the form of a supplement:
https://www.biogeosciences-discuss.net/bg-2019-323/bg-2019-323-AC4-supplement.pdf

---

## Author Comment (AC5) · 4 Dec 2019

The comment was uploaded in the form of a supplement:
https://www.biogeosciences-discuss.net/bg-2019-323/bg-2019-323-AC5-supplement.pdf